# The prolactin receptor scaffolds Janus kinase 2 via co-structure formation with phosphoinositide-4,5-bisphosphate

**Raul Araya-Secchi[1,2], Katrine Bugge[3†], Pernille Seiffert[3†], Amalie Petry[4], Gitte W Haxholm[3], Kresten Lindorff-Larsen[3], Stine Falsig Pedersen[4]\*, Lise Arleth[1]\*, Birthe B Kragelund[3]\***

[1]Structural Biophysics, Section for Neutron and X-ray Science, Niels Bohr Institute, University of Copenhagen, Copenhagen, Denmark; [2]Facultad de Ingenieria Arquitectura y Diseño, Universidad San Sebastian, Santiago, Chile; [3]Structural Biology and NMR Laboratory (SBiNLab), Department of Biology, University of Copenhagen, Copenhagen, Denmark; [4]Section for Cell Biology and Physiology, Department of Biology, University of Copenhagen, Copenhagen, Denmark

**\*For correspondence:**
sfpedersen@bio.ku.dk (SFP);
arleth@nbi.ku.dk (LA);
bbk@bio.ku.dk (BBK)

[†]These authors contributed equally to this work

**Competing interest:** The authors declare that no competing interests exist.

**Abstract** Class 1 cytokine receptors transmit signals through the membrane by a single transmembrane helix to an intrinsically disordered cytoplasmic domain that lacks kinase activity. While specific binding to phosphoinositides has been reported for the prolactin receptor (PRLR), the role of lipids in PRLR signaling is unclear. Using an integrative approach combining nuclear magnetic resonance spectroscopy, cellular signaling experiments, computational modeling, and simulation, we demonstrate co-structure formation of the disordered intracellular domain of the human PRLR, the membrane constituent phosphoinositide-4,5-bisphosphate ($PI(4,5)P_2$) and the FERM-SH2 domain of the Janus kinase 2 (JAK2). We find that the complex leads to accumulation of $PI(4,5)P_2$ at the transmembrane helix interface and that the mutation of residues identified to interact specifically with $PI(4,5)P_2$ negatively affects PRLR-mediated activation of signal transducer and activator of transcription 5 (STAT5). Facilitated by co-structure formation, the membrane-proximal disordered region arranges into an extended structure. We suggest that the co-structure formed between PRLR, JAK2, and $PI(4,5)P_2$ locks the juxtamembrane disordered domain of the PRLR in an extended structure, enabling signal relay from the extracellular to the intracellular domain upon ligand binding. We find that the co-structure exists in different states which we speculate could be relevant for turning signaling on and off. Similar co-structures may be relevant for other non-receptor tyrosine kinases and their receptors.

## Editor's evaluation

This important interdisciplinary study substantially advances our understanding of the prolactin receptor interactions with the membrane lipids and the effect of these interactions on cell signaling. The authors use a combination of state-of-the-art NMR structural analysis, simulations, and cellular assays to provide compelling experimental evidence for protein complexes being regulated by IDR-membrane interactions. The work will be of broad interest to structural biologists and biochemists, and the results presented herein are likely relevant for other non-tyrosine kinase receptors.

## Introduction

Cytokine receptors are transmembrane glycoproteins that bind cytokines on the cell surface and transduce signals across the membrane to the interior of the cell. This, in turn, activates signaling pathways leading to multiple outcomes including induction of immune responses, cell proliferation,

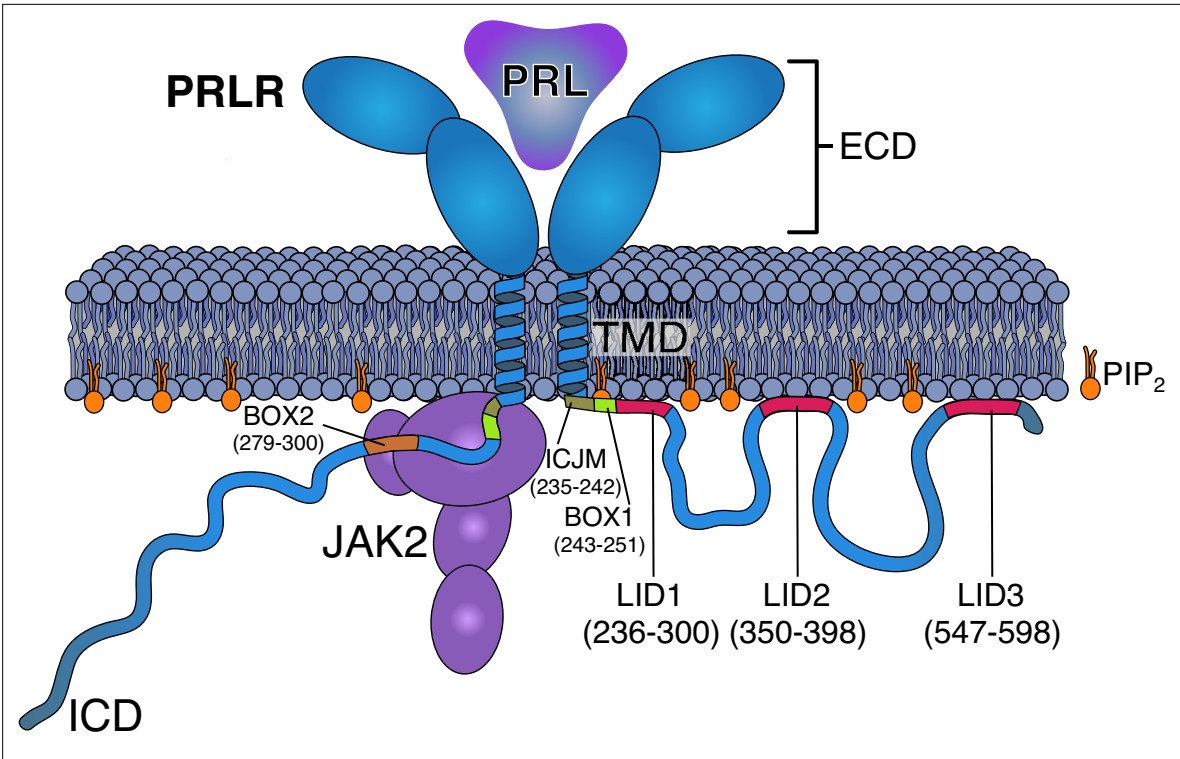

**Figure 1.** Schematics of the PRLR:PRL:JAK2 complex in the membrane. The PRLR is shown in light blue, the PRL as a dark blue triangle, the PRLR-ICD as a disordered chain, and JAK2 in purple. The PI(4,5)P$_2$ lipid (PIP2) is shown in orange. The intracellular juxtamembrane (ICJM) region and BOX1 of PRLR-ICD are highlighted in green nuances, while the three LIDs as defined in *Haxholm et al., 2015* are highlighted in red. For simplicity, only one of the two ICDs is shown associated with JAK2 via the BOX1 (green) and BOX2 (orange) motifs. PRLR, prolactin receptor; ICD, intracellular domain; LID, lipid interaction domains.

altered metabolism, and differentiation (*Brooks et al., 2016*). Class 1 cytokine receptors constitute a subclass of receptors that transverse the membrane by a single α-helical transmembrane domain (TMD; *Brooks et al., 2016*), separating a folded extracellular domain (ECD) from a disordered intracellular domain (ICD). The prolactin (PRL) receptor (PRLR) is one of the structurally most simple cytokine receptors (*Figure 1*). Signaling by the PRLR/PRL complex is implicated in the regulation of more than 300 biological functions in vertebrates (*Bole-Feysot et al., 1998*), and its function is especially well-known for its essential role in mammary gland development and lactation (*Hannan et al., 2023*). Apart from this, deregulation of PRLR/PRL signaling is associated with several pathologies in humans of which hyperprolactinemia resulting in reproductive failure is best described (*Bachelot and Binart, 2007*; *Newey et al., 2013*). Deregulation of PRLR/PRL signaling is linked to other diseases and, although still debated, suggested to be implicated in the development and progression of prostate (*Sackmann-Sala and Goffin, 2015*) and breast (*Clevenger and Rui, 2022*) cancers.

For cytokine receptors, signal transduction through the membrane is received by an ICD, which is intrinsically disordered and lacks kinase activity (*Haxholm et al., 2015*). Thus, association of auxiliary kinases is mandatory for signaling, with the Janus kinases (JAK1–3 and TYK2) being the most thoroughly described (*Brooks et al., 2016*; *Morris et al., 2018*). A proline-rich region constituting the so-called BOX1 motif close to the membrane, as well as a second hydrophobic motif termed BOX2, are known to be essential for JAK binding (*Figure 1*; *Ferrao et al., 2018*; *Rowlinson et al., 2008*). However, although progress has been made in the molecular understanding of cytokine binding and despite several structures of folded ECDs (*Broutin et al., 2010*; *de Vos et al., 1992*), TMDs (*Bocharov et al., 2018*; *Bugge et al., 2016*), a complete receptor (*Kassem et al., 2021*), and a receptor ICD in complex with JAK1 (*Glassman et al., 2022*) have emerged, it is still not clear how the signal inside the cell is received by the disordered region to elicit and control signal transduction.

A subset of class 1 cytokine receptors form homodimers and trimeric complexes with their ligands, with the main dimerization occurring in the TMDs (*Brooks et al., 2014*; *Brown et al., 2005*; *Gadd*

*and Clevenger, 2006*; *Kubatzky et al., 2001*; *Seubert et al., 2003*). This group includes the growth hormone receptor (GHR), the erythropoietin receptor, the thrombopoietin receptor, and the PRLR, which have become well-established paradigmatic models. Recently, signal transduction by the GHR has been suggested to occur via a rotation of the transmembrane helices within the dimer leading to a subsequent separation of the ICDs (*Brooks et al., 2014*; *Brown et al., 2005*). A torque is hereby exerted on the associated JAK2s, which is thought to relieve inhibition by the pseudokinase domains, initiating signaling. The ICDs of these receptors have been shown to be highly disordered (*Haxholm et al., 2015*), a feature which is preserved in models of the PRLR (*Bugge et al., 2016*) and the full-length GHR in nanodiscs (*Kassem et al., 2021*). This brings forward the question of how signaling is orchestrated by disorder and how a disordered linker region between the TMD and the region bound to the kinases can communicate and transduce information.

For both the PRLR and the GHR, lipid interaction domains (LID) with affinity for negatively charged lipids have been identified in their ICDs (*Haxholm et al., 2015*). Common to both receptors is that they contain a LID proximal to the membrane, directly overlapping with the JAK2 interaction sites, BOX1 and BOX2 (*Seiffert et al., 2020*). Using nuclear magnetic resonance (NMR) spectroscopy, we identified three LIDs in the PRLR-ICD termed LID1, LID2, and LID3 (*Figure 1*; *Haxholm et al., 2015*), and using lipid dot-blots, we showed that the PRLR-ICD has variable affinities for different membrane constituents, including for different phosphoinositides (PIs). In particular, PRLR has a distinct affinity for PI-4,5-bisphosphate (PI[4,5]P$_2$) and lacks affinity for PI(3,4,5)P$_3$ (*Haxholm et al., 2015*). PIs are important constituents of the membrane and play key roles in signaling, both as membrane inter-action partners that can be specifically modulated by phosphorylation (*Carracedo and Pandolfi, 2008*) and as secondary messengers (*McLaughlin et al., 2002*; *Suh and Hille, 2005*). Indeed, some single-pass receptors contain conserved anionic lipid interaction sites close to the membrane (*Hedger et al., 2015*), and increasing evidence suggests lipid interaction to take on important regulatory roles (*McLaughlin et al., 2005*; *Metcalf et al., 2010*). Recently, the epidermal growth factor receptor (EGFR) was shown to sequester PI(4,5)P$_2$ by accumulating it around the TMD regulating the dimer/monomer equilibrium and with a possible positive feedback loop through the activation of the phospholipase C (PLC) – diacylglycerol (DAG)-IP$_3$ pathways (*Maeda et al., 2018*). This will lead to subsequent conversion of PI(4,5)P$_2$ to PI(3,4,5)P$_3$ and hence depletion of PI(4,5)P$_2$ from the membrane. Similar depletion of PI(4,5)P$_2$ from the plasma membrane has been noted under hypoxia (*Lu et al., 2022*). For class 1 cytokine receptors, the role of PIs in signaling is less clear.

In a cellular context, signaling-related proteins can be membrane anchored through modifications such as acylation (*Patwardhan and Resh, 2010*; *Rawat et al., 2013*; *Rawat and Nagaraj, 2010*) or via designated lipid-binding domains. This includes the four point-1, ezrin, radixin moesin (FERM) domain of radixin, focal adhesion kinase (FAK) and the protein tyrosine phosphatase L1 (PTPL1; *Bompard et al., 2003*; *Feng et al., 2015*; *Hamada et al., 2000*), the SH2 domains of the Src family kinases (*Park et al., 2016*; *Sheng et al., 2016*), and the FERM-SH2 domain of merlin (*Mani et al., 2011*). Thus, colocalization of receptors and related signaling proteins can occur at the plasma membrane without necessarily being bound within a complex. It is, however, unclear whether such membrane co-localization has functional consequences, such as enhancing signaling speed, and whether the membrane may act as an additional scaffolding platform that enhances binding via restriction in the two-dimensional plane.

Recent work on disorder in membrane proteins and on the interplay between membrane proteins and lipids has revealed the need for strong integrative methods, combining successfully various structural biology techniques, biophysics, and computational biology (*Basak et al., 2022*; *Larsen et al., 2022*). These include NMR, small-angle X-ray scattering, crosslinking-mass spectrometry, and single molecule fluorescence combined with molecular dynamics (MD) simulations (*Chavent et al., 2018*; *Goretzki et al., 2023*). These efforts have provided important insights into the role of lipids in regulation of membrane proteins. For TRPV4, a member of the TRP vanilloid channel family, it was shown that an autoinhibitory patch of the receptor competed with PI(4,5)P$_2$ binding at the membrane to attenuate channel activity, and MD simulation showed that lipid binding affected the ensemble dynamics (*Goretzki et al., 2023*). For EphA2, a receptor tyrosine kinase, an integrative study showed how PIs mediate the interaction between the kinase domains, facilitated by clustering of PIPs via binding to the receptor juxtamembrane domain (*Chavent et al., 2018*), further promoting conformation specific dimerization (*Stefanski et al., 2021*). Thus, studying dynamic processes at the membrane

interface is an emerging field requiring integrative structural biology approaches for detailed atomic resolution information.

For the PRLR, it is still not clear whether, and if so how, interactions between the ICD and the membrane impact signal transduction and association with JAK2. Nor is it understood how structural disorder can relay and transmit information from the TMD to initiate signaling. To shed light on the molecular details underlying a potential interplay between the receptor, membrane, and kinase, we focused on the human PRLR and its LID1 closest to the membrane, facilitating the first intracellular step in signaling. Using an integrative approach combining NMR spectroscopy, cell biology, and computational modeling, we demonstrate the formation of a co-structure composed of the disordered PRLR-ICD, the membrane constituent PI(4,5)P$_2$, and the FERM-SH2 domain of the JAK2. Facilitated by this co-structure, the disordered region closest to the membrane forms an extended structure, which we suggest stabilizes the disordered domain, allowing signal relay from the extracellular to the ICD.

## Results

### LID1 is disordered in solution and when tethered to the transmembrane helix

Membrane interactions by PRLR-ICD have previously been studied in the absence of anchoring to the TMD defining three LIDs, with LID1 closest to the membrane (*Haxholm et al., 2015*). Since tethering would increase the local concentrations at the membrane and the ICD, this could affect affinity, complex lifetime as well as the degree by which structure formation would be captured. Furthermore, the first LID, LID1, is disordered and is located in the juxtamembrane region where it is responsible for transmitting information on extracellular hormone binding to the bound JAK2. As this constitutes the very first step on the intracellular side and given the distance to the other two LIDs (LID2 and LID3) and their disconnect from the TMD by long disordered regions, we focused exclusively on LID1. We recombinantly expressed the TMD (residues 211–235 with numbering corresponding to the processed protein) with five residues added at the two termini (TMD$_{F206-V240}$), and the TMD with the first 35 residues of LID1, TMD-ICD$_{F206-S270}$ (*Figure 2A*). We then examined their structural propensities in detergents and in small unilamellar vesicles (SUVs) by NMR spectroscopy. In 1,2-dihexanoyl -sn-glycero-3-phosphocholine (DHPC) micelles, most of the TMD resonances of the TMD$_{F206-V240}$ and TMD-ICD$_{F206-S270}$ variants were readily superimposable in the $^{15}$N,$^1$H-HSQC spectra (*Figure 2—figure supplement 1*), suggesting that the conformation of the TMD was not affected by the presence of the ICD. For the TMD-ICD$_{F206-S270}$, C$^α$ NMR resonances were collected for most of the disordered region, while backbone carbon resonances for the TMD, except for A222, and the region G236-P246 immediately following it were broadened beyond detection in the 3D spectra, preventing assignments (*Figure 2B*). This may suggest that the first 10 residues of the ICD interact with, are buried in the micelles, or are affected by the overall slower tumbling of the micelle, whereas the complete overlap of the TMD residues in the $^{15}$N-HSQC spectra confirm the helical structure as seen previously. We assigned the backbone nuclei of the detectable resonances of TMD-ICD$_{F206-S270}$ in DHPC micelles and compared the secondary chemical shifts (SCSs) to those of the ICD alone (ICD$_{G236-Q396}$; *Figure 2C*). Whereas the region of the ICD that is undetected in TMD-ICD$_{F206-S270}$ formed transient extended structures in the absence of the TMD, the observable parts were highly similar suggesting lack of structure induction by TMD tethering. In 1-palmitoyl-2-oleoyl-sn-glycero-3-phosphocholine (POPC) SUVs, only the resonances of the most C-terminal residues of TMD-ICD$_{F206-S270}$ were detectable; however, the chemical shifts suggested that the residues were disordered (*Figure 2—figure supplement 2*). Taken together, these results suggest that most of the ICD residues remain disordered when tethered to the TMD and in the presence of a neutral membrane mimetic.

### LID1 binds PI(4,5)P$_2$ in the juxtamembrane region forming extended structures

PRLR-ICD has previously been shown to bind PI(4,5)P$_2$ (but not PI[3,4,5]P$_3$; *Haxholm et al., 2015*), suggesting that this lipid could modulate membrane affinity and the structural properties of the PRLR-ICD. To separate headgroup effects from lipid bilayer surface effects, we used a short-chain C$_8$-PI(4,5)P$_2$, which has a high critical micelle concentration (CMC) of 2 mM (*Goñi et al., 2014*) and

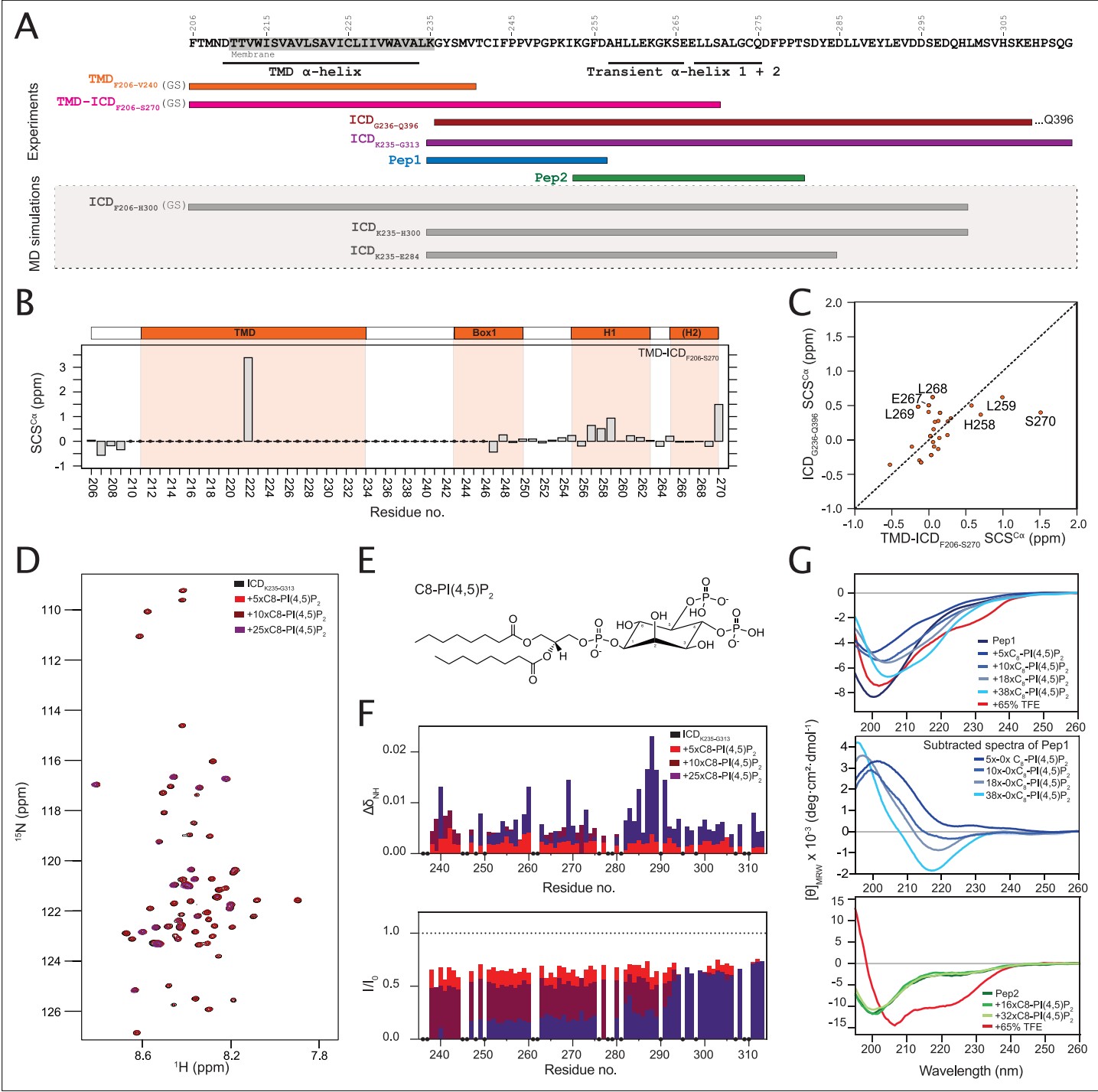

**Figure 2.** The intracellular juxtamembrane (ICJM) region of the prolactin receptor (PRLR) interacts with PI(4,5)P$_2$. (**A**) Overview of investigated PRLR variants. (**B**) Secondary chemical shifts (SCSs) of transmembrane domain (TMD)-intracellular domain (ICD)$_{F206-S270}$ reconstituted in 1,2-dihexanoyl-sn-glycero-3-phosphocholine (DHPC) micelles. (**C**) Correlation plot of the SCSs of ICD$_{G236-Q396}$ plotted against those of TMD-ICD$_{F206-S270}$. (**D**) $^{15}$N,$^1$H-HSQC spectra of $^{15}$N-ICD$_{K235-G313}$ titrated with 5×, 10×, and 25× molar excess of C$_8$-PI(4,5)P$_2$. (**E**) Structure of C$_8$-PI(4,5)P$_2$. (**F**) Backbone amide chemical shift perturbations (CSPs) and peak intensity changes upon addition of C$_8$-PI(4,5)P$_2$ to $^{15}$N-ICD$_{K235-G313}$ plotted against residue number. (**G**) Top: Far-UV CD spectra of Pep1 titrated with C$_8$-PI(4,5)P$_2$ or in 65% trifluoroethanol (TFE). Middle: Far-UV CD spectra of Pep1 in the presence of 5 x-38x C$_8$-PI(4,5)P$_2$ subtracted with the spectrum of Pep1 in the absence of C$_8$-PI(4,5)P$_2$. Bottom: Far-UV CD spectra of Pep2 titrated with C$_8$-PI(4,5)P$_2$ or in 65% TFE.

The online version of this article includes the following figure supplement(s) for figure 2:

**Figure supplement 1.** $^{15}$N, $^1$H-HSQC spectra.

**Figure supplement 2.** C$^\alpha$ secondary chemical shifts of intracellular domain (ICD)$_{G236-Q396}$.

analyzed the structural changes by NMR and CD spectroscopy at concentrations below the CMC (*Goñi et al., 2014*) to identify the binding site.

$^{15}$N-labeled ICD$_{K235-G313}$ covering LID1 (*Figure 1*) was titrated with C$_8$-PI(4,5)P$_2$, and binding was assessed by $^1$H-$^{15}$N-HSQC spectra (*Figure 2D–F*). The chemical shift perturbations (CSPs) were modest, whereas substantial intensity changes were observed throughout the chain, supporting a direct interaction between the ICD and lipids. The resonances from G236 to F244 completely disappeared suggesting exchange on an intermediate NMR timescale, while intensities were substantially reduced in the V247–S290 region (*Figure 2F*). In the region from D285 to E292, we observed an almost inverse correlation between the CSPs and the intensities. This suggests that in contrast to the preceding region, a faster local exchange rate allows us to follow the resonances from the bound state in this region, giving rise to the large CSPs. From this region and to the C-terminus, only moderate intensity changes were observed (*Figure 2F*). These findings suggest that the primary PI(4,5)P$_2$ binding site is located closest to the membrane in what we define as the intracellular juxtamembrane (ICJM) region (K235-C242). The ICJM is located N-terminally to the BOX1 motif ($_{243}$IFPPVPGPK$_{251}$ [UNIPROT]; $_{245}$PPVPGPK$_{251}$ [http://elm.eu.org/]).

As the resonance-broadening precluded observation of the bound state, two overlapping peptides, Pep1 (K235-D256) and Pep2 (K253-T280), were constructed and evaluated by CD spectroscopy. In isolation, the peptides were disordered as judged by the negative ellipticity at 200 nm in their far-UV CD spectra (*Figure 2G*). Pep1 and Pep2 were titrated with C$_8$-PI(4,5)P$_2$ and the structural changes monitored (*Figure 2G*). For Pep2, the far-UV CD signal was unaffected by the presence of C$_8$-PI(4,5)P$_2$. In contrast, for Pep1, large spectral changes were seen, which were unrelated to helix formation. Subtracting the spectra in the presence and absence of C$_8$-PI(4,5)P$_2$ revealed a negative ellipticity minimum at 218 nm, a strong indicator of β-strands, showing that when bound to C$_8$-PI(4,5)P$_2$, a distinct extended (strand-like structure) signature was seen (*Figure 2G*). This suggests that this region of LID1 changes its conformational properties upon binding to C$_8$-PI(4,5)P$_2$. We evaluated the intrinsic helical propensities of the two ICD segments by exposure to high trifluorethanol (TFE) concentrations. Here, Pep2 was readily able to form helical structure as expected from the presence of two transient helices (*Haxholm et al., 2015*; *Figure 2G*, *top*), whereas Pep1 was not (*Figure 2G*, *bottom*).

In summary, LID1 of the PRLR-ICD interacts with PI(4,5)P$_2$, with the primary interaction site located in the K235–S290 region. Headgroup interaction was dominantly located to the region K235–D256 constituting the ICJM and the BOX1 motif and this interaction induced the formation of a regional extended structure in the PRLR-ICD.

## LID1 has specific PI(4,5)P$_2$ contacts which drive PI(4,5)P$_2$ recruitment

To obtain a more detailed characterization of the behavior of the disordered PRLR-ICD near lipid bilayers, as well as the effect of different anionic lipid headgroups in these, we turned to molecular simulations. Here, as explained above, we focused on the LID1 (K235–H300), alone and in context of TMD, and first placed a coarse-grained (CG) model of the TMD-ICD$_{F206-H300}$ in three different mixed-membrane systems. These contained an upper leaflet consisting of 100% POPC and lower leaflets composed of a 90:5:5 or 80:10:10 mixture of POPC:POPS:PI(4,5)P$_2$ (*Figure 3AB*) or a 70:30 molar ratio mixture of POPC:1-palmitoyl-2-oleoyl-sn-glycero-3-phospho-L-serine (POPS; *Figure 3—figure supplement 1E, F*). Since the Martini force field may produce unrealistically collapsed disordered regions, increasing the strength of the protein-water interactions by 8–10% has provided satisfactory results when applied to the simulation of other disordered regions or multidomain proteins (*Thomasen et al., 2022*). Thus, the simulations were run using a modified version of the Martini3 force field with a 10% increase in the strength of the protein–water interactions. For comparison, similar simulations were performed using the Martini2 force field (*Figure 3—figure supplement 1*).

We first analyzed the dynamics of the LID1 during the simulations focusing on the pattern of protein-lipid contacts. Here, we measured the number of protein-lipid contacts focusing either on interactions between the protein and lipid headgroups or the protein and lipid acyl chains. In both cases, we determined the fraction of the simulation time that the protein and different parts of the lipid were within 7 Å of each other. In general, we observed that residues in the N-terminal part of the LID1 (K235 - D255), which includes the ICJM and BOX1, established contacts with the bilayer in all three membrane systems (*Figure 3A and B*, *Figure 3—figure supplement 2*). Furthermore, a hydrophobic region rich in prolines (V240–P250) made consistent contacts with the acyl-chains and much

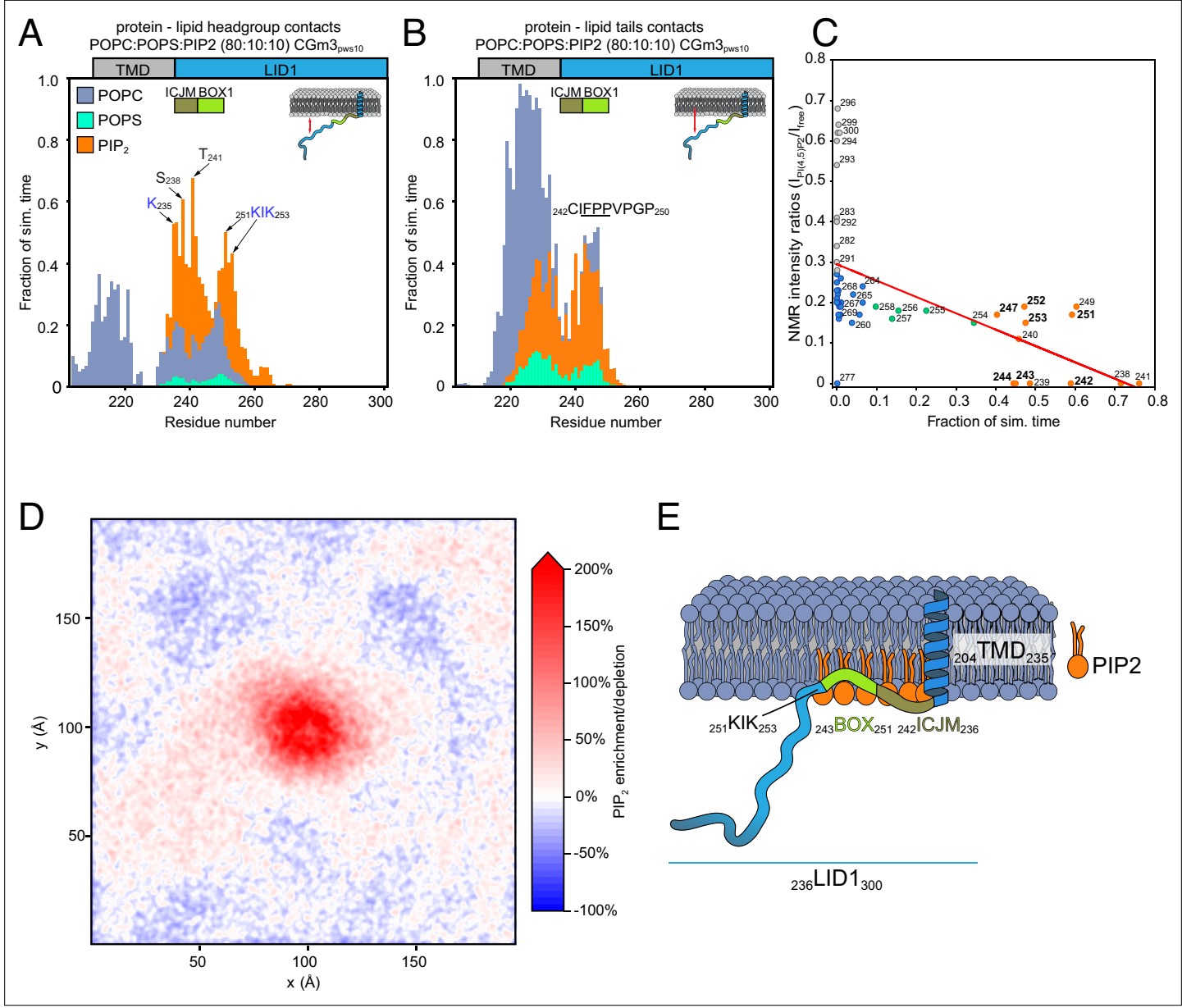

**Figure 3.** Protein–lipid interactions of prolactin receptor (PRLR)-intracellular domain (ICD)$_{LID1}$ obtained from coarse-grained (CG)-molecular dynamic (MD) simulations. (**A–B**) Protein–lipids contact histograms for PRLR-ICD$_{LID1}$ +POPC:POPS:PI(4,5)P$_2$ (80:10:10). (**A**) Contacts between the protein and lipid headgroups. A contact is counted if the distance between the backbone beads of the protein is ≤7 Å from the head-group beads of the lipids. (**B**) Contacts between the protein and the acyl chains of the lipids. A contact is counted if the distance between the backbone bead of the protein is ≤7 Å from the acyl-chain bead of the lipids. (**C**) Correlation between the change in nuclear magnetic resonance (NMR) signal and the contact frequency between PRLR-ICD$_{LID1}$ and the lipid headgroups from the PRLR-ICD$_{LID1}$ +POPC:POPS:PI(4,5)P$_2$ (80:10:10) system. Pearson correlation coefficient of –0.55 with p=4.0 × 10$^{-5}$ and $R^2$=0.3. (**D**) Average PI(4,5)P$_2$ density map (*xy*-plane) taken from the PRLR-ICD$_{LID1}$ +POPC:POPS:PI(4,5)P$_2$ (80:10:10) simulation. The map is colored according to the enrichment/depletion percentage with respect to the average density value. (**E**) Schematic representation of how the interactions and the embedment into the membrane of PRLR contribute to the co-structure formation. The data from the simulations correspond to those of the production stage (see methods). POPC, 1-palmitoyl-2-oleoyl-sn-glycero-3-phosphocholine; POPS, 1-palmitoyl-2-oleoyl-sn-glycero-3-phospho-L-serine.

The online version of this article includes the following figure supplement(s) for figure 3:

**Figure supplement 1.** Protein–lipid interactions of prolactin receptor (PRLR)-intracellular domain (ICD)$_{LID1}$ obtained from coarse-grained (CG)-molecular dynamic (MD) simulations using the martini 2.2 force field.

**Figure supplement 2.** Complementary analysis of protein–lipid interactions of prolactin receptor (PRLR)-intracellular domain (ICD)$_{LID1}$ obtained from coarse-grained (CG)-molecular dynamic (MD) simulations using the martini 3.0b3.2 force field.

more than to the headgroups, indicating penetration into the lower-leaflet. Similar behavior has been reported with CG-MD simulations for the juxtamembrane region of other single-pass transmembrane receptors (*Hedger et al., 2015*). For PRLR, the pattern of interaction was independent on the lipid composition, at least in terms of protein-POPC contacts, and the region interacting with the lipids was similar in all three membrane systems (*Figure 3A and B*, *Figure 3—figure supplement 2*).

Although the extent and pattern of protein–lipid interactions appeared constant, a striking observation was made in both systems containing PI(4,5)P$_2$. Here, protein–lipid interactions between residues K235 and K253 and PI(4,5)P$_2$ were present during a large fraction (≥50%) of the simulations, despite PI(4,5)P$_2$ being present at only 5 or 10% of the total lipids (*Figure 3—figure supplement 2*). This was also observed in simulations with the Martini2 force field, in which the LID1 promptly collapsed in a globular and unstructured coil (*Figure 3—figure supplement 1*). This suggested that PI(4,5)P$_2$ spontaneously accumulated, or in other ways became concentrated around the TMD-ICD$_{F206-H300}$. The computed average density maps for PI(4,5)P$_2$ indeed showed that PI(4,5)P$_2$ formed a microdomain around the TMD (*Figure 3D*). The low frequency of contacts between the protein and POPS suggests that POPS did not accumulate or compete with PI(4,5)P$_2$ for binding to the TMD-ICD$_{F206-H300}$, further supporting the preference for PI(4,5)P$_2$ observed earlier (*Haxholm et al., 2015*). Similar preference was also observed with 5% PI(4,5)P$_2$ (*Figure 3—figure supplement 2*) as well as with Martini2 (*Figure 3—figure supplement 2E–F*).

A more detailed analysis of LID1-PI(4,5)P$_2$ contacts revealed a preference for certain residues, shown as peaks in the protein-headgroup contact profiles. In particular, K235, S238, T241, K251, and K253, which define a KIK motif suggested as a PI(4,5)P$_2$ binding motif (*Kjaergaard and Kragelund, 2017*), engaged in highly populated contacts (*Figure 3A* and *Figure 3—figure supplement 2A*). The pattern of contacts was not affected by PI(4,5)P$_2$ concentration; however, the frequency of contacts almost doubled as a result of the increase from 5 to 10% of PI(4,5)P$_2$. The hydrophobic residues in the ICJM and BOX1 penetrate the headgroup layer and facilitate the approximation of the KIK motif to the PI(4,5)P$_2$ headgroups. The stabilization of the structure provided by the hydrophobic residues from the ICJM and BOX1 is also reflected in their decrease in flexibility, observed as a shoulder on the backbone RMSF plot, for the residues that comprise the ICJM and BOX1 of PRLR-LID1. Very similar profiles of the backbone RMSF plot were obtained for the systems with respectively 5 and 10% PI(4,5)P$_2$ in POPC:POPS, suggesting that the loss in flexibility is coupled to the buried hydrophobic residues rather than to specific PI(4,5)P$_2$ interaction (*Figure 3—figure supplement 2C*). Contributions from other positively charged residues such as K262 and K264 were very small. To validate the observations from the simulations, we compared the pattern of protein:PI(4,5)P$_2$ interactions observed in the NMR experiments to those from the simulations containing PI(4,5)P$_2$ (*Figure 3C*). A clear correlation between loss of NMR signal and a high amount of protein-PI(4,5)P$_2$ and POPC/POPS contacts in the 80:10:10 simulation was observed, reinforcing that the simulations are able to capture the specificity of protein-PI(4,5)P$_2$ interactions. Furthermore, both experiments (*Figure 2*) and simulations (*Figure 3*) show that the residues involved in binding to PI(4,5)P$_2$-containing membranes overlap with those that are involved in binding to JAK2.

In summary, the CG-simulations of TMD-ICD$_{F206-H300}$ near lipid membranes showed accumulation of PI(4,5)P$_2$ around the TMD and the N-terminual part of LID1 involving the ICJM and BOX1. The residues made contact with the membrane independently of lipid type, with BOX1 residues acting as a tether by penetrating the headgroups. This tethering keeps positively charged residues, such as K251 and K253, near the membrane. This may be the driver for PI(4,5)P$_2$ recruitment, enhanced by higher PI(4,5)P$_2$ concentration. Intriguingly, we observed that the same regions involved in JAK2 binding (BOX1) also play roles in membrane association and lipid recruitment.

## JAK2-FERM-SH2 and PRLR-ICD$_{LID1}$ form co-structures with PI(4,5)P$_2$ on membranes

It has been suggested that JAK2 and PRLR interact constitutively in cells (*Campbell et al., 1994*; *Rui et al., 1994*), although recent data for the GHR have shown that the Src family kinase Lyn competes for this site (*Chhabra et al., 2023*). Thus, given our observations that residues from LID1 form lipid-specific contacts with the membrane constituents using the same region covering the binding interface with the FERM domain of JAK2, we decided to explore the structure and dynamics of the JAK2(FERM-SH2):PRLR(LID1) complex near lipid bilayers (*Figure 4*). To do so, an atomistic model of the complex

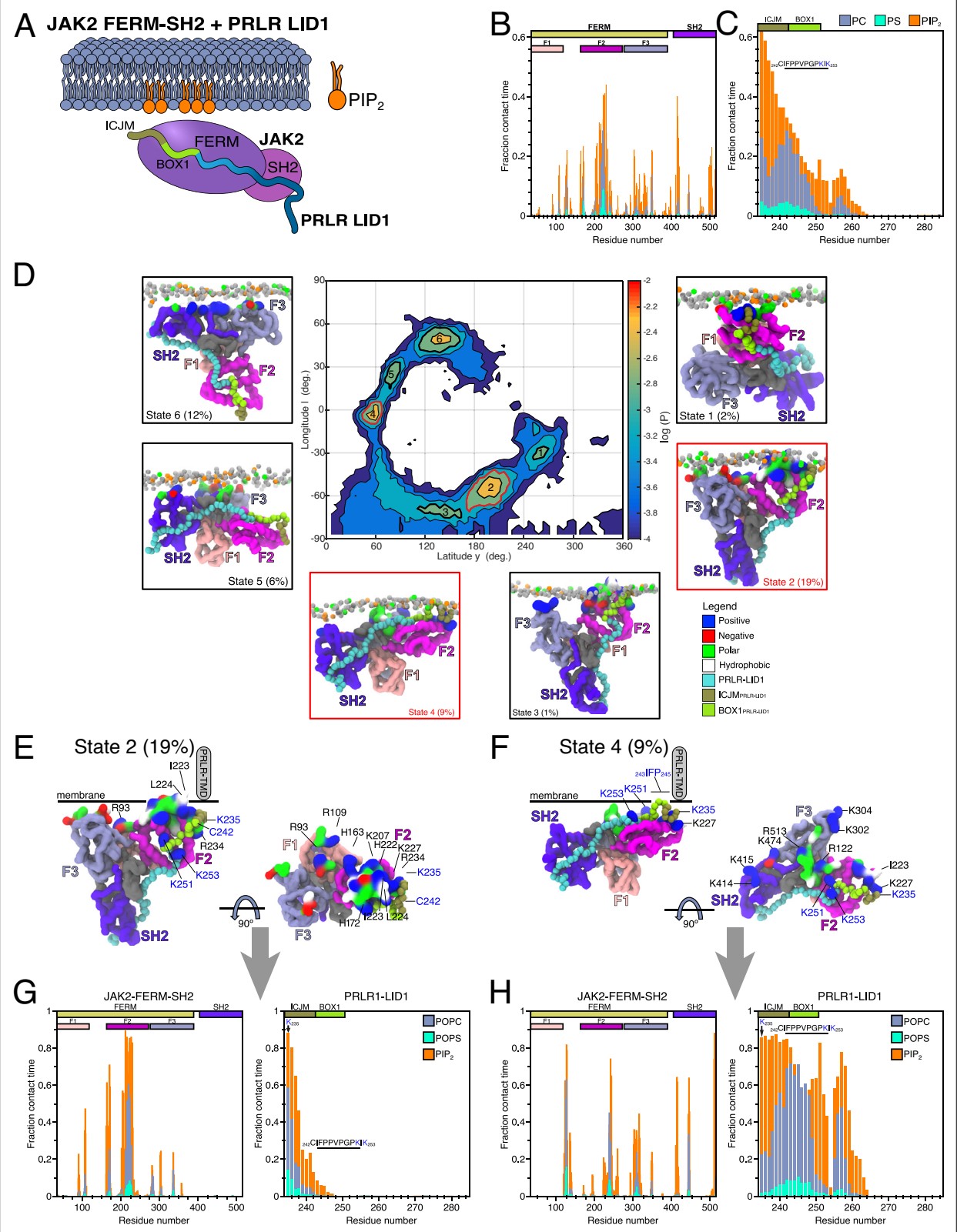

**Figure 4.** Protein–lipid interactions of the JAK2-FERM-SH2 PRLR-ICD$_{LID1}$ complex obtained from coarse-grained (CG)-molecular dynamic (MD) simulations. (**A**) Schematic representation of the simulated system. Combined (**B**) JAK2-FERM-SH2-lipid and (**C**) PRLR-ICD$_{LID1}$-lipid contact frequency histograms for the 16 CG simulations of the JAK2-FERM-SH2+PRLR-ICD$_{LID1}$+POPC:POPS:PI(4,5)P$_2$ system. (**D**) Distribution of the orientations adopted by the JAK2-FERM-SH2+PRLR-ICD$_{LID1}$ complex when bound to lipids taken from the 16 simulations with POPC:POPS:PI(4,5)P$_2$ in the lower-leaflet. The

*Figure 4 continued on next page*

*Figure 4 continued*

snapshots surrounding the map correspond to representative conformations of the highlighted states also indicating the fraction total bound time for which each state was observed. Representative conformations of (**E**) State 2 and (**F**) State 4. The gray cylinder depicts the position where PRLR-transmembrane domain (TMD) should be located. Representative protein–lipid contact histograms for (**G**) State 2 and (**H**) State 4 colored as in panels B and C. POPC, 1-palmitoyl-2-oleoyl-sn-glycero-3-phosphocholine; POPS, 1-palmitoyl-2-oleoyl-sn-glycero-3-phospho-L-serine.

The online version of this article includes the following video and figure supplement(s) for figure 4:

**Figure supplement 1.** Analysis of the JAK2-FERM-SH2- PRLR-ICD$_{K235-H300}$ all atom MD simulation.

**Figure supplement 2.** Complementary analysis of protein–lipid interactions of the JAK2-FERM-SH2 PRLR-intracellular domain (ICD)$_{K235-H300}$ complex obtained from coarse-grained (CG)-molecular dynamic (MD) simulations.

**Figure supplement 3.** Snapshots of the different binding states observed for the JAK2-FERM-SH2 – prolactin receptor (PRLR)-intracellular domain (ICD)$_{K235-H300}$ complex with the complete structural model of JAK2 (obtained from AF2-EBI database).

**Figure 4—video 1.** Y State (STATE 2) from the JAK2-FERM-SH2 PRLR-intracellular domain (ICD)$_{K235-E284}$ complex simulated near a bilayer containing PI(4,5)P$_2$: Representative trajectory showing State 2 (Y).
https://elifesciences.org/articles/84645/figures#fig4video1

**Figure 4—video 2.** FLAT State (STATE 4) from the JAK2-FERM-SH2 PRLR-intracellular domain (ICD)$_{K235-E284}$ complex simulated near a bilayer containing PI(4,5)P$_2$: Representative trajectory showing State 4 (Y).
https://elifesciences.org/articles/84645/figures#fig4video2

---

of a smaller region of PRLR-ICD$_{K235-E284}$ bound to the JAK2-FERM-SH2 domains (residues P37 to T514) was built, taking advantage of crystal structures of JAK2-FERM-SH2 and of JAK1-FERM-SH2 and TYK2-FERM-SH2 bound to analogous fragments of the ICDs of the interferon $\lambda$ - and α-receptors (IFNLR1 and IFNAR1), respectively. This model was used to perform all-atom MD simulations in a water-box to obtain equilibrated structures for further simulations and to study the dynamics of the protein complex. The average contact map between JAK2-FERM-SH2$_{P37-T514}$ and PRLR-ICD$_{K235-E284}$ showed clusters of contacts in which residues from BOX1 of LID1 formed close contacts (avg. dist. ≤4 Å) with residues from the F2 lobe (and the F1–F2 linker) and the SH2 domain of JAK2-FERM-SH2, respectively (*Figure 4—figure supplement 1A*). C-terminally to BOX1, a second set of persistent contacts was observed, again involving charged and hydrophobic residues including F255, L259, E261, K262, and K264 from PRLR-ICD$_{K235-E284}$. Conservation analysis using ConSurf (*Ashkenazy et al., 2016*; *Landau et al., 2005*; *Figure 4—figure supplement 1B–D*) showed conserved residues in the interface, particularly those of BOX1 of PRLR (P245, P248, K251, and I252; *Figure 4—figure supplement 1B*), while a patch of conserved residues (T225, I229, and F236) in JAK2-FERM-SH2 formed close contact with residues from PRLR BOX1. JAK2 residues V183 and L184 interacted with the backbone of $_{251}$KIK$_{253}$ of PRLR-ICD, whereas other, less conserved residues such as E173 and E177 formed transient salt-bridges with K251 and K253 of PRLR-ICD (see *Figure 4—figure supplement 1A*). The contact map also showed that the N-terminus of the ICJM remained flexible without close contacts with JAK2-FERM-SH2 (avg. dist. ≥6 Å).

Next, an equilibrated structure of the JAK2-FERM-SH2:PRLR-ICD$_{K235-E284}$ complex was used to build a CG model, which was then placed near lipid bilayers of different lower leaflet composition. A number of randomly positioned starting orientations was placed ~7 Å below the lower leaflet (16 orientations for the POPC:POPS:PI[4,5]P$_2$ [80:10:10] membrane and eight for the POPC:POPS [70:30] membrane). In addition, we included 12 orientations of JAK-FERM-SH2 without the PRLR-ICD$_{K235-E284}$ placed near a POPC:POPS:PI(4,5)P$_2$ (80:10:10) membrane. For the JAK2-FERM-SH2:PRLR-ICD$_{K235-E284}$ complex near a 70:30 POPC:POPS membrane, binding to the lower leaflet was observed for only three of the eight systems (*Figure 4—figure supplement 2A*). In contrast, when PI(4,5)P$_2$ was present (10%), rapid binding of the complex to the membrane was observed in all simulations reaching 97% saturation (*Figure 4—figure supplement 2A*). Both proteins in the complex showed specific clusters of residues with contacts to PI(4,5)P$_2$, POPS, and POPC, independent of the initial orientations (*Figure 4BC*). The number of contacts formed was higher for the simulations with PI(4,5)P$_2$. This suggests that contacts with other components of the membrane occur close to the bound PI(4,5)P$_2$. Overall, the PRLR-ICD$_{K235-E284}$ showed a pattern of lipid contacts similar to the simulations of PRLR TMD-ICD$_{G204-H300}$ with the POPC:POPS:PI(4,5)P$_2$ (80:10:10) membrane (*Figure 4C*), with residues K$_{235}$GY$_{237}$ contacting PI(4,5)P$_2$ headgroups for at least 50% of the total contact time and with insertion into the membrane; note that this occurs even though PRLR-ICD$_{K235-E284}$ is not tethered to the membrane via the TMD. Also,

like in the PRLR TMD-ICD$_{G204-H300}$ simulations, contacts made by C242 and I243 to POPC were still present. In contrast, contacts by other residues from BOX1 and the KIK motif have lower populations. However, and as expected from the location of the most frequent PRLR-ICD $_{K235-E284}$/lipid contacts, JAK2-FERM-SH2 had more contacts in the F2 lobe of the FERM domain, mainly involving residues I223, L224, R226, K227, and R230, constituting the regions where the N-terminus of PRLR-ICD$_{K235-E284}$ is bound (*Figure 4B*). In the JAK2-FERM-SH2 simulations without PRLR-ICD and near a POPC:POPS:PI(4,5)P$_2$ (80:80:10) membrane, we observed that simulations initiated in 11 out of the 12 orientations ended up binding to the membrane (*Figure 4—figure supplement 2A*). In this case, the overall binding pattern of protein-lipid contacts remained similar.

Previous studies have suggested that the Martini2 force field model underestimates cation-π interactions between surface aromatic residues and choline headgroups on the membrane (*Khan et al., 2020*). However, this may not be applicable to other types of protein–membrane interactions, particularly where negatively charged headgroups are present. Our simulations involving PI(4,5)P$_2$ and POPS suggest that interactions between PRLR and the bilayer are primarily driven by positively charged residues in the protein and that other protein–membrane interactions are secondary or occur between the lipids and residues that surround positively charged residues interacting with a PI(4,5)P$_2$ (or POPS) lipid. As a result, cation-π interactions may not be as important for the protein–lipid contact patterns we observed but could be one explanation as to why we observe less frequent binding in the POPC:POPS systems.

In summary, our simulations showed that binding of JAK2 to the membrane was enhanced by the presence of PI(4,5)P$_2$ and that the ICD from PRLR and JAK2 formed a co-structure with PI(4,5)P$_2$ maintaining the contacts to the lipids observed for the individual proteins. The presence of PI(4,5)P$_2$ was essential for the membrane interactions.

## The complex between JAK2-FERM-SH2 and LID1 shows preferential bound orientations with membranes containing PI(4,5)P$_2$

To characterize the membrane-bound modes of the complex in more detail, we took inspiration from Vogel et al. (*Herzog et al., 2017*) and constructed a map that represents the populations of different orientations of the JAK2-PRLR-ICD$_{K235-E284}$ complex relative to the membrane and extracted conformations to represent the most populated orientations as classified into states (*Figure 4D*). For the JAK2-PRLR-ICD$_{K235-E284}$ complex bound to the POPC:POPS:PI(4,5)P$_2$ (80:10:10) membrane, states 1–4 (~31% of the total contact time) showed the complex in an orientation where the N-terminus of PRLR-ICD$_{K235-E284}$ contacted and inserted into the bilayer similarly to what was observed in the PRLR TMD-ICD$_{G204-H300}$ simulations near a POPC:POPS:PI(4,5)P$_2$ bilayer. Of these four states, states 1, 2, and 3 had the F2 lobe of the JAK2-FERM domain and the ICJM region of PRLR-ICD$_{K235-E284}$ in contact with the membrane, penetrating below the headgroups, and acting as a pivot over which the protein-complex rotates, leaving the complex to hang as a 'Y' from the membrane (see *Figure 4DE* and *Figure 4—video 1*). State 4 on the other hand, while retaining the main contact points, assumed a 'flat' orientation with larger sections of the F2 lobe and F1–F2 linker from JAK2-FERM and the entire N-terminal half of PRLR-ICD$_{K235-E284}$ (residues K235-G263) making a substantial number of contacts with the membrane (*Figure 4DF* and *Figure 4—video 2*). To examine whether the identified states are compatible with functional states of the full-length kinase, we superimposed representative conformations of states 1–6 with that of the full-length JAK2 model obtained from the AlphaFold Protein Structure Database (UNIPROT O60674; *Jumper et al., 2021*; *Varadi et al., 2022*). This procedure revealed that both the Y (states 1, 2, and 3), and Flat (state 4) states keep the other domains of JAK2 oriented toward the cytoplasmic space (*Figure 4—figure supplement 3A*), supporting that these states could be functionally relevant. Furthermore, in the context of JAK2 dimerization required for signaling (*Ferrao et al., 2018*), these states allow for the correct orientation for kinase domain dimerization. The two remaining states (states 5 and 6) showed an inverted orientation in which the main protein–lipid interactions were formed by residues from the F3 lobe of FERM and the SH2 domain, bringing the F2 lobe and the ICJM region of the PRLR-ICD unrealistically far away from the membrane and from the connecting end of the TMD. Thus, states 5 and 6 appear functionally irrelevant, as further demonstrated by the superposition of the full-length AlphaFold model of JAK2 in which the kinase domains would clash with the bilayer (*Figure 4—figure supplement 3E, F*).

## Different membrane co-structures have different exposures relevant to signaling

We analyzed the protein–lipid interactions formed by states 2 (Y) and 4 (Flat) in more detail, and despite overall similar contact profiles, some key differences were observed (*Figure 4E–H*). For the Flat state, an increase in contacts was seen for residues K235 to L260 of PRLR-ICD$_{K235-E284}$ with a pattern similar to the one observed in the PRLR-TMD-ICD$_{G204-H300}$ simulations with PI(4,5)P$_2$ present. Dominant PI(4,5)P$_2$ contacts were seen for K235–C242, followed by POPC contacts for residues C242–P248, with a second PI(4,5)P$_2$ contact peak for K251 and a third around H257. For the Y state, only residues K235–Y237 made substantial contacts with PI(4,5)P$_2$ and/or POPC, leaving BOX1 and the KIK motif exposed to the solvent and making contacts with JAK2. For JAK2-FERM-SH2, the main difference between the Y and Flat states was a large decrease in contacts to the F2 lobe in the Flat state accompanied by an increase in contacts in F3 and SH2. Residues in the F2 lobe involved in homodimerization, orientation, and activation, including L224, K227, R230, and R234 (*Wilmes et al., 2020*), were only accessible in the Y state and not in the Flat state. Thus, we speculate that the Y and Flat states may mimic functionally relevant conformations pertaining to active and inactive states of the signaling complex. Mapping of residues that make contacts with the bilayer in the two states to the conservation maps shows that several positively charged residues of JAK2-FERM-SH2 are largely conserved. Particularly, K207, R226, K227, and R228 in F2 with high population contacts with PI(4,5)P$_2$ in the Y state are highly conserved. Similar conservation was seen for the positively charged residues K235, K414, K415, and R513 that form contacts with PI(4,5)P$_2$ in the Flat state.

Similar simulations were performed near a POPC:POPS 70:30 bilayer, which resulted in only one of eight systems showing stable binding to the bilayer characterized by only one state (*Figure 4—figure supplement 2A and B*). Here, residues from the F2 lobe of JAK2 form contacts with POPS lipids in a narrow peak containing residues Q219–R230, while for PRLR-ICD $_{K235-E284}$, residues from the ICJM (K235–C242), and the BOX1 region (C242–P248) make high-frequency contacts with both POPS and POPC (*Figure 4—figure supplement 2D*). Overall, this state is somewhat similar to state 3 observed for the simulations with PI(4,5)P$_2$. Similarly, for our simulations of JAK2-FERM-SH2 without PRLR near a bilayer with PI(4,5)P$_2$, 11 out of 12 stable binding conformations revealed three most populated states (E1, E2, and E3; *Figure 4—figure supplement 2C*). Characterization of these in terms of protein–lipid contact profiles revealed that the positively charged residues in the F2 lobe play an important role in binding to PI(4,5)P$_2$ in a similar manner as observed for the Y and Flat states of the complex (*Figure 4—figure supplement 2F*). Indeed, states 2 and 3 (*Figure 4—figure supplement 2G*) show remarkable similarities with the Flat and Y states from the simulations of the JAK2-FERM-SH2:PRLR-ICD$_{K235-E284}$ complex near a similar bilayer.

Overall, the simulations highlight preferential binding of both JAK2-FERM-SH2, both alone and in complex with PRLR-ICD$_{K235-E284}$ to PI(4,5)P$_2$, and show that the absence of this lipid decreases the level of LID1 binding to the bilayer. Even in the absence of TMD tethering, the most populated bound states recapitulate the binding mode observed for the PRLR-ICD alone. Another key observation is the existence of different states in which different regions of both JAK2 FERM-SH2 domain and LID1 of PRLR are exposed to the solvent or hidden below the bilayer.

## Key residues for membrane interaction control cellular signaling efficiency

From the NMR experiments and MD simulations, we identified residues in LID1 that interact with different components of the membrane and/or the FERM-SH2 domain of JAK2. This resulted in four clusters positioned in the ICJM (K235–C242), the BOX1 region (C242–P248), two basic patches of the KxK motif type (K251–K253 and K262–K264), and hydrophobic residues in the region connecting them, respectively. To decipher the specific role of these clusters for PI(4,5)P$_2$ interaction, we introduced four sets of mutations in ICD$_{K235-G313}$ and investigated the effect on PI(4,5)P$_2$ interaction using NMR spectroscopy. Based on the NMR data and simulations, we focused on the KxK motifs, which would be involved in binding to PI(4,5)P$_2$ (*Figure 3A*) and JAK2 (*Figure 4—figure supplement 1*), the CIF sequence, indicated to be important to membrane binding (*Figures 2 and 3B*) and four hydrophobic residues, where at least two were seen to be important for JAK2 binding (*Figure 4—figure supplement 1*). We avoided interfering directly with the BOX1 core motif (P245–P250; *Pezet et al., 1997*). Thus, the CIF motif (C242–F244) was mutated to GAG (GAG mutant: C242G, I243A,

and F244G), while the lysines in the KxK motifs (251–253 and 262–264) were all mutated to either glycines (K4G mutant: K251G, K253G, K262G, and K264G) or, for charge reversal, to glutamates (K4E mutant: K251E, K253E, K262E, and K264E). Finally, four hydrophobic residues ($I_{252}$, $F_{255}$, $L_{259}$, and $L_{260}$) were mutated to glycines (φ4G mutant). $^{15}N$-PRLR-ICD$_{K235-G313}$ and the four variants were titrated with up to 25×molar excess of $C_8$-PI(4,5)$P_2$, keeping the concentration below the CMC. $^1H$-$^{15}N$-HSQC spectra were recorded at each titration point, and the changes in chemical shifts and signal intensities were quantified (*Figure 5A*). Similarly to wild type (WT), all variants showed negligible chemical shift changes (*Figure 5—figure supplement 1*) but large peak intensity changes. Decreased peak intensities were observed for all variants in the region of G236–D295, where the changes were largest for the φ4G mutant and K4G, and similar to WT, while smaller effects were seen for the GAG and the K4E variants, suggesting weaker affinities. Together, this indicates that the KxK motifs and the CIF-motif are involved in PI(4,5)$P_2$ interaction, as expected from the contacts predicted from simulation and NMR and CD data, yet none of these residues are essential for binding.

We observed dramatic increases in peak intensities for the K4G and φ4G variants in the presence of 5× and 10× molar excess of $C_8$-PI(4,5)$P_2$ when compared to WT (*Figure 5A*), suggesting changes in the dynamics of the chain. To address this, we probed the backbone dynamics by acquiring $^{15}N$ $R_2$ relaxation rates of the WT, K4G, and φ4G variants in the absence of $C_8$-PI(4,5)$P_2$ (*Figure 5—figure supplement 2*). Compared to the WT, no major changes in $R_2$ were observed for two variants. The intensity increase observed for the K4G and φ4G variants during the titration with $C_8$-PI(4,5)$P_2$ therefore indicates increased backbone dynamics upon binding to $C_8$-PI(4,5)$P_2$ compared to the WT. Although we cannot explain this observation, it suggests that binding to $C_8$-PI(4,5)$P_2$ increases the dynamics of the first parts of the chain and thus require higher concentration of $C_8$-PI(4,5)$P_2$ to fully form the complex, as expected by the lower apparent affinity.

To address the implications of PI(4,5)$P_2$ interaction for PRLR membrane localization and downstream signaling and to enable a potential separation of effect of perturbed membrane localization from direct PI(4,5)$P_2$ binding, we introduced the same four sets of mutations into the full-length PRLR. Together with WT PRLR, these were transiently transfected into AP1 mammalian epithelial cells, which were stably transfected with the fluorescent PI(4,5)$P_2$ reporter 2PH-PLCδ-GFP. The cells were subjected to fluorescence microscopy analysis of PRLR and the PI(4,5)$P_2$ reporter (*Figure 5B*) and to western blot analysis of STAT5-activating phosphorylation (*Figure 5C–D and F–G*). None of the mutations fully abolished PRLR membrane localization. Western blot analysis showed that the protein expression levels were similar for WT PRLR and all PRLR variants (*Figure 5C*). However, compared to WT, the K4G, K4E, and φ4G variants exhibited a significant reduction in membrane localization as determined by line scan analysis (*Figure 5B and E*). This is in accordance with their predicted JAK2 contacts obtained from the simulations, as JAK2 is known to be important for PRLR trafficking (*Huang et al., 2001*). The PRL-induced STAT5 activation was significantly decreased in cells expressing either the K4G, K4E, or the φ4G variants, whereas STAT5 activation in cells expressing the GAG variant was not significantly different from that of WT expressing cells (*Figure 5C and F*). Decomposing the K4E variant into two individual mutants, in which only one of the two KxK motifs was changed (K2E$_{251}$ and K2E$_{262}$), showed that the reduction in STAT5 activation was attenuated in the variants with the individual mutations, compared to the drastic decrease observed for the K4E mutant (*Figure 5D and G*). Thus, both KxK motifs are important for JAK/STAT activation, which suggests that both PI(4,5)$P_2$ and JAK2 binding are important in this regard.

Taken together, these results show that while our data are consistent with the decreased membrane localization contributing to the reduction of STAT5 activation, it is unlikely to account fully for the effect observed for the K4G, K4E, or the φ4G variants. Part of this reflects impaired binding to JAK2, known to affect the amount of receptor at the cellular membrane. The MD simulations indicated that only the first KxK motif is involved in lipid interaction while the second KxK motif is involved in JAK2 interaction. Thus, a part of the the reduction in JAK/STAT activation in these variants could arise from a combined effect of abolishing both PI(4,5)$P_2$- and JAK2 interaction within the LID1 region, which support the suggestion that co-structure formation between JAK2, PRLR, and the membrane is critical for optimal JAK/STAT signaling. However, within this co-structure, the involved residues will likely affect several binding events, and thus, separation of function by selective mutations may not be straightforward.

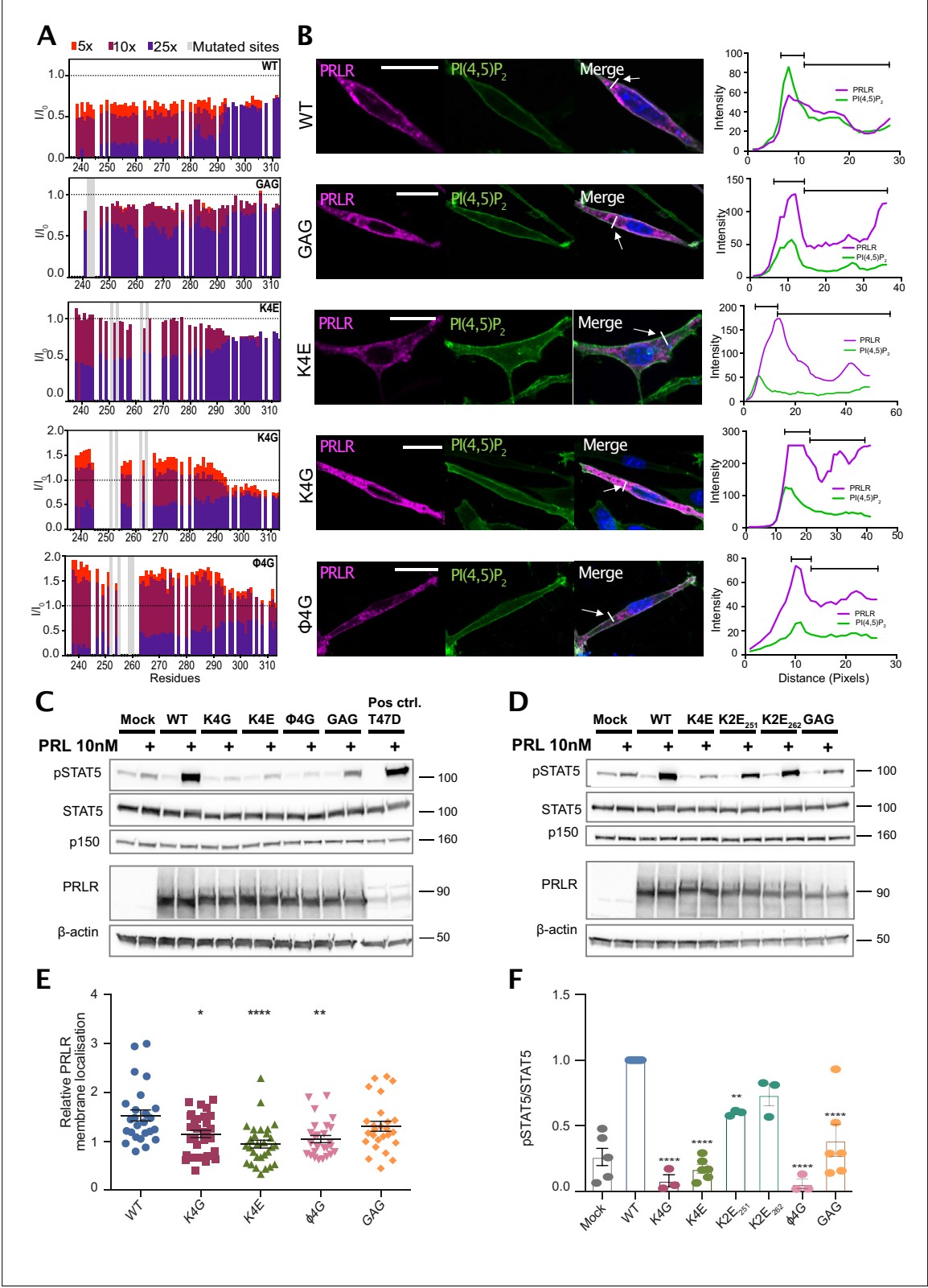

**Figure 5.** PRLR variants with mutations in lipid interacting residues exhibit decreased PRL-stimulated STAT5 activation in AP1-2PH-PLCδ-GFP cells. (**A**) NMR peak intensity changes of PRLR-ICD$_{K235-G313}$ variants: WT, K4G, K4E, φ4G, and GAG variants upon titration with 5×, 10×, and 25× molar excess C$_8$-PI(4,5)P$_2$ plotted against residue number. (**B**) The PRLR variants (WT, K4G, K4E, φ4G, and 3GAG) were transiently transfected in AP1 cells stably expressing the 2PH-PLCδ-GFP construct which visualizes the plasma membrane by binding PI(4,5)P$_2$. The cells were subsequently analyzed

*Figure 5 continued on next page*

*Figure 5 continued*

by immunofluorescence microscopy, using antibodies against PRLR (magenta) and GFP (green), as well as DAPI (blue) to mark nuclei. To the right, examples of an average line-scan for each PRLR variant are shown. The fluorescence intensity depicted along the white line drawn (arrow) and green fluorescence (plasma membrane) was used to divide the line in a plasma membrane section and intracellular section, and relative membrane localization was calculated as the average fluorescence of PRLR in the membrane section divided by that in the intracellular section. (**C and D**) AP1-2PH-PLCδ-GFP cells were transiently transfected with PRLR variants (WT, K4G, K4E, φ4G, 3GAG, K2E$_{253}$, and K2E$_{261}$), incubated overnight followed by serum starvation for 16–17 h and were subsequently incubated with or without 10 nM prolactin for 30 min. The resulting lysates were analyzed by western blot for STAT5, pSTAT5 (Y964), PRLR, with β-actin and p150 levels. The western blots shown are representative of three biological replicates. (**E**) Ratio of plasma membrane localized receptor compared to intracellular receptor, analyzed by line-scans as in B. Each point represents an individual cell, and data are based on three independent biological experiments per condition. Graphs show means with SEM error bars. *p<0.05, **p<0.01, and ****p<0.0001. One-way ANOVA compared to WT, unpaired. (**F**) Quantification of western blot results shown as pSTAT5 normalized to total STAT5, relative to the WT condition. Graphs show means with SEM error bars. *p<0.05 and **p<0.01. One-way ANOVA compared to WT, unpaired.

The online version of this article includes the following source data and figure supplement(s) for figure 5:

**Source data 1.** Raw western blots (relating to *Figure 5C*).

**Source data 2.** Raw western blot (relating to *Figure 5D*).

**Source data 3.** Data summaries (relating to *Figure 5E and F*).

**Figure supplement 1.** Chemical shift perturbations of intracellular domain (ICD)$_{K235-G313}$ of (**A**) WT (**B**) K4E, (**C**) GAG, (**D**) K4G, and (**E**) φ4G variants.

**Figure supplement 2.** $^{15}$N $R_2$ relaxation rates of intracellular domain (ICD)$_{K235-G313}$ of WT (gray bars), K4G (blue dots) and φ4G (red squares) variants.

## Discussion

The sequence of the human PRLR has been known for more than 35 years (*Boutin et al., 1988*). Little attention has, however, been given to the role of membrane composition for PRLR signaling, despite it being placed in the plasma membrane where PI levels are highly dynamic and spatially variable and being linked to cancer with lipid deregulation (*Dadhich and Kapoor, 2022*). Here, we asked if JAK2 and PRLR-ICD share a PI(4,5)P$_2$ binding site and if and how the binding to PI(4,5)P$_2$ plays a role in the orientation of these proteins with respect to the membrane. Integrating MD simulations with biophysical and cellular experiments has been critical in this endeavor. Our first goal was to identify the residues of LID1 involved, as well as the structure formed—if any—in the protein–lipid complex. Our results suggest that the residues that form the ICJM and BOX1 regions of the ICD interact with the lipids via non-specific hydrophobic interactions that involve penetration of the bilayer below the headgroups. This in turn enables positively charged residues of the $_{251}$KIK$_{253}$ motif to establish ionic interactions with PI(4,5)P$_2$, and in doing so the region folds into an extended structure, similar to structures of other cytokine receptors in complex with either JAK1 or TYK2 (*Wallweber et al., 2014*; *Zhang et al., 2016*). In turn, PI(4,5)P$_2$ lipids accumulate around the TMD and LID1 of PRLR, suggesting a relevant functional role of the interaction.

The results highlight the capacity of the LID1 to establish highly populated and specific interactions with PI(4,5)P$_2$ via residues essential for its interaction with JAK2. Therefore, we addressed whether LID1 in complex with the FERM-SH2 domain of JAK2 could engage with the lipid bilayer in the absence of the TMD and with or without PI(4,5)P$_2$ lipids. Indeed, PI(4,5)P$_2$ was required for binding of the complex to the membrane, as the presence of only POPS in the lower leaflet was not enough to sustain binding despite its negative charge. Remarkably, when PI(4,5)P$_2$ was present, we observed specific binding orientations that positioned the ICJM region of the LID1 in the same position as when tethered to the TMD. Furthermore, in the complex, the protein–lipid contact profiles were similar to the one observed for LID1 alone suggesting the PI(4,5)P$_2$ binding pattern to be maintained in complex with JAK2. A detailed study of two of the most populated PRLR-bound states of JAK2 revealed a striking difference in orientation and contact pattern with the lipids that could shed light on functionally relevant states. For example, the most populated state, the Y state, had contacts from the F2 lobe of the JAK2-FERM-SH2 domain and the ICJM of the PRLR, which penetrated the membrane forming hydrophobic interactions with the acyl chains. In this orientation, regions of both JAK2 and the PRLR that have been associated with receptor dimerization and activation for signaling (*Ferrao et al., 2018*; *Wilmes et al., 2020*) are exposed to solvent and available for interactions. We note that this orientation has resemblance to that shown in recent cryo electron microscopy (EM) structures of JAK1 bound to IFNAR1 (*Glassman et al., 2022*). For the Flat state, we observed a drastic change in the protein–lipid contact profiles for both proteins but more markedly for LID1. While the main interaction

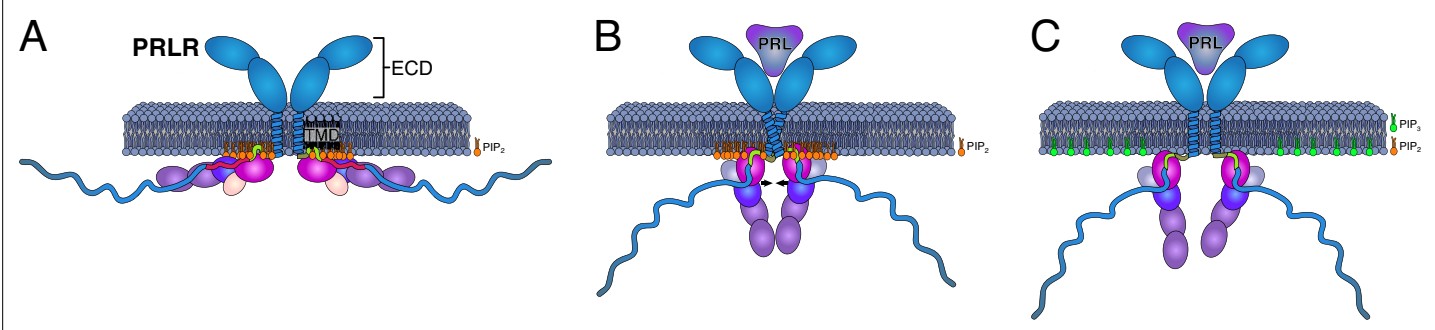

**Figure 6.** Model of how co-structure formation between JAK2, prolactin receptor (PRLR), and PI4(4,5)P₂ may contribute to signaling fidelity. The suggested states in signaling would be (**A**) the inactive state of the co-structure exemplified by the Flat orientation. (**B**) The hormone bound state expemplified by the co-structure in the Y orientation. (**C**) Phosphorylation of PIP(4,5)P₂ to PI(3,4,5)P₃ for which the PRLR has no affinity may lead to downregulation and/or termination of signaling. The color scheme of the proteins is identical to *Figure 4*.

site remains the F2 lobe of JAK2 and the ICJM of the PRLR, the N-terminal residues of the LID1 now lie sandwiched between the membrane and JAK2-FERM-SH2 domain and recapitulates the binding pattern observed from TMD-ICD$_{F206-H300}$ simulations with membranes containing PI(4,5)P₂. Remarkably, in this Flat state, most of the accessible regions in the Y states are now hidden under the membrane. Thus, we speculate that the Y and Flat states may mimic an available and hidden state, respectively, that could be relevant for regulation of dimerization and activation of signaling. Interestingly, one residue that has been suggested to play a major role in JAK2 membrane association, orientation, dimerization, and activation is L224 (*Wilmes et al., 2020*). This residue anchors into the membrane only in the Y state and not in the Flat state. Similarly, in simulations where L224 was mutated to glutamate, a change in preferred orientation together with loss of dimerization and JAK2 and STAT5 phosphorylation was observed (*Wilmes et al., 2020*). This further highlight that the orientation of the PRLR-JAK2 complex relative to the membrane has functional relevance. Importantly, the presence of PI(4,5)P₂ in the membrane structurally tightens the path from the ECD, folding the ICJM and BOX1 in an extended structure, making transmission of information of hormone binding possible. Thus, signal relay by disordered linker of the PRLR can now be possible through its complex formation with the membrane and JAK2 (*Figure 6*).

Lastly, we tested the functional relevance of our observations by mutating residues that appeared significant for PI(4,5)P₂:PRLR:JAK2 co-structure formation and determining their impact on cellular PRL signaling. Both KxK motifs and the hydrophobic residues connecting them were important for PI(4,5)P₂ interaction, PRLR membrane localization, and cellular JAK2/STAT5 signaling. From the MD simulation, it was however evident that not all residues in these motifs were in direct contact with the membrane, further highlighting that co-structure formation between PRLR, JAK2, and the membrane is essential for optimal signal transduction. Another interesting observation was that even though mutating the CIF motif had the largest impact on PI(4,5)P₂ binding, it had only a limited effect on cellular JAK2/STAT5 signaling. Since the NMR results suggested that the ICJM serves as a primary PI(4,5)P₂ anchoring point facilitating additional contacts along the chain, this could indicate that a cooperative interaction within the co-structure is needed to control signaling and that PI(4,5)P₂ interaction is necessary for proper and substantial co-structure formation.

The PI(4,5)P₂-specific interactions observed point toward a possible regulatory role of PI(4,5)P₂ in PRLR signaling. Our simulations showed that the membrane embedded TMD-ICD$_{F206-H300}$ was associated with an accumulation of PI(4,5)P₂ around the TMD. One of the suggested roles of PI(4,5)P₂ as a regulatory lipid is indeed to form microdomains around proteins and reduce their lateral movement (*Trimble and Grinstein, 2015*; *van den Bogaart et al., 2011*). Another possible role can be inferred from previous studies on the EGFR. Evidence suggests a positive feedback loop where inhibition is released upon activation because PI(4,5)P₂ is hydrolyzed to DAG and IP₃ by PLC$_\gamma$ (*Maeda et al., 2018*; *McLaughlin et al., 2005*). Specifically, we have previously shown that PRLR does not interact with PI(3,4,5)P₃ (*Haxholm et al., 2015*). As PI(4,5)P₂ is phosphorylated by the PI3-kinase to PI(3,4,5)P₃ during PRLR signaling (*Aksamitiene et al., 2011*; *Yamauchi et al., 1998*), this could indicate a way

of attenuating signaling. Whether hydrolysis of PI(4,5)P$_2$ by PLC$_\gamma$ is relevant for PRLR signaling is not known.

## Conclusions

Signal transduction by single-pass receptors through the membrane is still an enigma. In the present work, we identify co-structure formation of the disordered LID1 of the PRLR, the membrane constituent PI(4,5)P$_2$, and the FERM-SH2 domain of the JAK2 and demonstrate its importance for PRLR signaling. This co-structure has at least two orientations, a Y-shaped state extending from the membrane and a Flat state with sites hidden in the membrane, the functional roles of which await further elucidation. The co-structure led to accumulation of PI(4,5)P$_2$ at the TMD interface and mutation of residues identified to specifically interact with PI(4,5)P$_2$ negatively affected PRL-induced STAT5 activation. Facilitated by the co-structure, the disordered ICJM folds into an extended structure, tightening the path from the ECD to the ICD. We suggest that the co-structure formed between receptor, kinase, and PI(4,5)P$_2$ is critical for signal relay from the extracellular to the intracellular side of the membrane and that different orientations of the co-structure exist that may represent inactive and active states.

# Materials and methods
## Expression and purification of TMD $_{F206-V240}$, TMD-ICD $_{F206-S270}$, and ICD$_{G236-Q396}$

PRLR-ICD$_{G236-Q396}$ was produced as described in *Haxholm et al., 2015*, and TMD$_{F206-V240}$ and TMD-ICD$_{F206-S270}$ were produced as described in *Bugge et al., 2015*.

## Expression and purification of ICD$_{K235-G313}$ and variants (K4E, K4G, $\phi$4G, and GAG)

ICD$_{K235-G313}$ and variants hereof, K4E (K251E, K253E, K262E, and K264E), K4G (K251G, K253G, K262G, and K264G), $\phi$4G (I252G; F255G, L259G, and L260G), and GAG (C242G, I243A, and F244G) were all produced as follows: competent BL21(DE3) were transformed using heat shock transformation with pET24a+ plasmids encoding the protein of interest with N-terminal His$_6$-SUMO tag. One colony was used to inoculate 10 mL of Luria Bertani (LB) media with 50 µg/mL kanamycin and incubated overnight at 37°C at 160 RPM. The overnight culture was used to inoculate 1 L M9 minimal media (3 g/L KH$_2$PO$_4$, 7.5 g/L Na$_2$HPO$_4$, 5 g/L NaCl, 1 mM MgSO$_4$, 4 g/L glucose, 1 g $^{15}$NH$_4$Cl$_2$, 1 mL M2 trace solution, and 50 µg/mL kanamycin) and grown at 37°C. At OD600, ~0.6 recombinant protein expression was induced with 0.1 mM Isopropyl β- d-1-thiogalactopyranoside (IPTG) for 4 hr at 37°C. Cells were harvested by centrifugation (5000 × *g*, 20 min, 4°C) and kept at –20°C until purification. Cells were resuspended in 35 mL Buffer A (10 mM imidazole, 50 mM Tris [pH 8], 150 mM NaCl, and 2 mM dithiothreitol [DTT]) and lysed with French press at 25 kpsi, followed by centrifugation at 20.000 × *g*, 45 min, 4°C. The supernatant was applied to 5 mL of pre-equilibrated Ni-NTA beads and incubated for 15 min followed by 50 mL wash with Buffer B (10 mM imidazole, 50 mM Tris [pH 8], 1 M NaCl, and 2 mM DTT) and 50 mL wash with Buffer A. Protein was eluted with 10 mL Buffer C (250 mM imidazole, 50 mM Tris [pH 8], 150 mM NaCl, and 2 mM DTT). The elution was supplemented with 0.01 mg ULP-1 and dialyzed against 1 L of dialysis buffer (50 mM Tris [pH 8], 150 mM NaCl, and 1 mM DTT) overnight at 4°C. The sample was re-applied to the Ni-NTA column and incubated for 15 min. Flow through containing cleaved protein was collected, and the remaining protein was eluted with 10 mL Buffer C. The sample was supplemented with 10 mM DTT before heating at 75°C for 5 min with gentle rotation of the sample throughout. Sample was transferred directly to ice for 10–15 min incubation followed by centrifugation at 20.000 × *g*, 10 min, 4°C. The supernatant was concentrated and supplemented with 5 mM betamercoptoenthaonl (b-ME) before application to a HiLoad 16/60 Superdex75 prep grade column equilibrated in 20 mM Na$_2$HPO$_4$/NaH$_2$PO$_4$, 150 mM NaCl, 5 mM b-ME (pH 7.3). Fractions containing pure protein were pooled and concentrated.

## CD spectroscopy

The peptides covering residues K235–D256 (Pep1) and K253–T280 (Pep2), respectively, were purchased from KJ Ross (DK) at 95% purity from HPLC purification. The peptides were dissolved in 10 mM Na$_2$HPO4/Na$_2$HPO4, pH 7.3 to a final concentration of 40 µM (Pep1) and 25 µM (Pep2) and

titrated with TFE or $C_8$-PI(4,5)$P_2$. The spectra were recorded in a 1 mm Quartz cuvette on a Jasco-810 spectropolarimeter purged with 8 L/min $N_2$ at 25°C. A total of 10 accumulations were acquired from 260 to 190 nm with the following settings: 0.5 nm data pitch, 1 nm band width, response time of 2 s, scanning speed of 10 nm/min. A background reference was recorded at identical settings for each sample and subtracted from the relevant spectrum. The spectra were processed by fast Fourier transform filtering and ellipticity converted to mean residual ellipticity ($[\theta]_{MRW}$).

## NMR spectroscopy

### TMD$_{F206-V240}$ and TMD-ICD$_{F206-S270}$

$^{15}$N-labeled or $^{13}$C,$^{15}$N-labeled TMD-ICD$_{F206-S270}$ was solubilized in molar excess DHPC dissolved in 50 mM NaCl, 20 mM Na$_2$HPO$_4$/NaH$_2$PO4 buffer, pH 7.2. Subsequently, the DHPC embedded TMD-ICD$_{F206-S270}$ was subjected to thorough buffer exchange in a 3 kDa cutoff spinfilter to remove residuals. For reconstitution into POPC SUVs, $^{15}$N-labeled TMD-ICD$_{F206-S270}$ was solubilized in 300 µL 5:1 methanol:chloroform and mixed with molar excess POPC dissolved in chloroform. The constituents were mixed, followed by evaporation of the organic solvent under a stream of $N_2$. When the lipid film appeared dry, it was either left under a stream of $N_2$ or placed under vacuum for at least an hour. The resulting proteoliposome film was rehydrated with 1 mL of 50 mM NaCl, 20 mM Na$_2$HPO$_4$/NaH$_2$PO$_4$ buffer, pH 7.2, followed by extensive dialysis against the buffer in a 3.5 kDa MWCO dialysis tube at 4°C. Subsequently, the proteoliposome solution was sonicated in an ultrasonication bath or, if the solution did not clarify, with an UP400S Ultrasonic Processor, in rounds of 2 s with 30 s rest between runs. Finally, the sample was concentrated in a 3 kDa cutoff spinfilter.

All NMR samples of $^{15}$N-labeled or $^{13}$C,$^{15}$N-labeled TMD-ICD$_{F206-S270}$ were added 10% (v/v) D$_2$O, 2 mM tris(2-carboxyethyl)phosphine (TCEP), 1 mM sodium trimethylsilylpropanesulfonate (DSS), 0.05% (v/v) NaN$_3$, and 50 mM NaCl, 20 mM Na$_2$HPO$_4$/NaH$_2$PO4 buffer (pH 7.2) to a final volume of 370 µL followed by pH-adjustment to 7.2 (if needed). All spectra were acquired at 37°C because the peak intensities of the TMD region decreased at lower temperatures. Free induction decays were transformed and visualized in NMRPipe (*Delaglio et al., 1995*) and analyzed using the CcpNmr Analysis software (*Vranken et al., 2005*). Proton chemical shifts were referenced internally to DSS at 0.00 ppm, with heteronuclei referenced by relative gyromagnetic ratios. For assignments of backbone nuclei, heteronuclear NMR spectra of a sample containing 0.5 mM $^{13}$C,$^{15}$N-labeled TMD-ICD$_{F206-S270}$ in 500 times molar excess DHPC were acquired on a Bruker 750-MHz ($^1$H) equipped with a cryoprobe. HNCACB and CBCA(CO)NH spectra were acquired with 32 and 40 of transients, respectively, and 20% non-uniform sampling (*Mayzel et al., 2014*) and used for manual backbone assignments. SCSs were calculated using random coil chemical shifts from *Kjaergaard et al., 2011* (obtained by supplying primary structure, pH, and temperature to the webtool), which were subtracted from the assigned TMD-ICD$_{F206-S270}$ chemical shifts.

The $^1$H,$^{15}$N-HSQC spectrum of 0.4 mM $^{15}$N-labeled TMD-ICD$_{F206-S270}$ in POPC SUVs (100 times molar excess of POPC) was acquired on a Varian INOVA 750- MHz ($^1$H) spectrometer equipped with a room temperature probe. The number of transients was 104.

### ICD$_{K235-G313}$

ICD$_{K235-G313}$ and the four variants (K4E, K4G, phi4G, and GAG) were dialyzed at 4°C overnight against 20 mM Na$_2$HPO$_4$/NaH$_2$PO$_4$ (pH 7.3), 150 mM NaCl. The samples of 50 µM protein were added 1 mM TCEP, 0.25 mM DSS, and 10% (v/v) D$_2$O and centrifuged at 20.000 × $g$, 4°C for 10 min and transferred to 5 mm Shigemi BMS-3 tubes. All NMR experiments were recorded at 5°C on a Bruker Avance III 600 MHz ($^1$H) spectrometer equipped with cryogenic probe. Free induction decays were transformed and processed in qMDD (*Orekhov and Jaravine, 2011*), phased in NMRDraw (*Delaglio et al., 1995*), and analyzed in CcpNMR analysis software (*Vranken et al., 2005*). Proton chemical shifts were referenced to DSS and nitrogen and carbon to their relative gyromagnetic ratios. $^1$H-$^{15}$N-HSQC experiments were acquired using non-uniform sampling (*Mayzel et al., 2014*) and recorded on 50 µM $^{15}$N-ICD$_{K235-G313}$ (or variants) in the absence and presence of 5×, 10×, and 25× molar excess of $C_8$-PI(4,5)$P_2$ (Avanti Lipids 850185).

Transverse $^{15}$N relaxation rates ($R_2$) of ICD$_{K235-G313}$ and the two variants, K4G and φ4G, were acquired on Bruker Avance III 600 MHz ($^1$H) spectrometer with varying relaxation delays of 0 ms, 33.92 ms,

67.84 ms, 135.68 ms, 169.6 ms, 203.52 ms, 271.36 ms, and 339.2 ms, measured in triplicates and peak intensities fitted to single-exponential decays.

## Cell lines and media

AP1 cells were a kind gift from Prof. S. Grinstein, University of Toronto, Canada. The cell line originates from a Chinese Hamster Ovary cell line and were selected for lack of Na+/H+ exchanger activity (for details, see *Rotin and Grinstein, 1989*). The PI(4,5)P2 sensor 2PH-PLCδ-GFP (Addgene plasmid #35142) was stably expressed in the AP1 cells, resulting in the new cell line AP1-2PH-PLCδ-GFP, used in this study. AP1-2PH-PLCδ-GFP cells expressing the WT and mutant PRLR variants (WT, K4G, K4E, φ4G, and 3GAG) were grown in Minimum Essential Medium Eagle (EMEM, Gibco) containing 10% fetal bovine serum (Sigma Aldrich), 1% penicillin/streptomycin (Sigma), 1% L-glutamine (Sigma) and 600µg/mL geneticin (Merck-Millipore). Cell lines were maintained at 37°C with 95% humidity and 5% $CO_2$ and were passaged by gentle trypsination for a maximum of 15 passages. Their identity was routinely validated by $NH_4Cl$ prepulse experiments for pH regulation phenotype, western blotting for transporters, receptors, and signaling, and immunofluorescence analysis for 2PH-PLCδ-GFP. They are not on the list of commonly misidentified cell lines. Cells were routinely checked for mycoplasma every 2–3 months and were consistently mycoplasma free.

## Immunoblotting

Cells were grown to ~80% confluence in 6-well plates, washed in ice-cold PBS, lysed in boiling lysis buffer (1% sodium dodecyl sulfate (SDS), 10 mM Tris–HCl, pH 7.5, with phosphatase inhibitors), boiled for 1 min, sonicated, and centrifuged to clear debris. Identical amounts of protein (12 µg/well) diluted in NuPAGE LDS sample buffer (Novex) with 50% 0.5 M DTT were boiled for 5 min, separated on Bio-Rad 10% Tris-Glycine gels, and transferred to nitrocellulose membranes using the Trans-Blot Turbo Transfer system (Bio-Rad). Membranes were stained with Ponceau S to confirm equal loading, blocked for 1 hr at 37°C in blocking buffer (TRIS-buffered saline with 0.1% Tween-20 (TBST), 5% nonfat dry milk), and incubated with the relevant primary antibodies in blocking buffer overnight at 4°C. After washing in TBST (TBS +0.1% Tween-20), the membranes were incubated with Horseradish peroxidase (HRP)-conjugated secondary antibodies (1:2000, Sigma), washed in TBST, and visualized using enhanced chemiluminescence (ECL) reagent (Bio-Rad). Protein bands were quantified by densitometry using ImageJ software and normalized to those of STAT5 and then to WT.

## Immunofluorescence analysis

For immunofluorescence experiments, cells were grown on 12 mm round glass coverslips to ~80% confluency and fixed in 2% PFA (30 min at RT). Coverslips were washed three times for 3×5 min in PBS, permeabilized for 15 min (0.5% Triton X-100 in TBS), blocked for 30 min (5% BSA in TBST), and incubated with primary antibody in TBST +1% BSA at RT for 1.5 hr. Coverslips were again washed in TBST+1% BSA and incubated with AlexaFluor488 and AlexaFluor568 conjugated secondary antibody (1:600 in TBS +1% BSA) for 1.5 hr. Finally, coverslips were incubated with DAPI (1:1000) for 5 min to stain nuclei, washed in TBST, and mounted in N-propyl-gallate mounting medium (2% w/v in PBS/glycerol). Cells were visualized using the Olympus IX83 microscope with a Yokogawa spinning disc confocal unit, using a 60×/1.4 NA oil emersion objective. Image adjustments were carried out using ImageJ software. Line scans were performed using the ColorProfiler ImageJ software plugin.

## Primary antibodies

The following antibodies were used: PRLR (Santa Cruz #SC20992), STAT5 (Santa Cruz #SC835), pSTAT5 (Y964; Cell Signaling #CS4322), p150 (BD #BD610473), and β-actin (Sigma Aldrich #A5441).

## Modeling of simulated proteins

### PRLR TMD-LID1 on a lipid bilayer

To build a model of the hPRLR-TMD-ICD-LID1 region (G204–H300), we used the MODELLER interface of Chimera (*Pettersen et al., 2004*; *Webb and Sali, 2016*). The structure of hPRLR-TMD (PDB 2N7I [*Bugge et al., 2016*]) was used as template for the transmembrane helix (in this structure, the residue at position 204 [P] was mutated to a G thus, in our model position 204 corresponds to a glycine), and due to the lack of structural templates for the ICD, it was modeled as a disordered

coil. This all-atom model was used to build CG simulation systems where the TMD was embedded in different lipid bilayers composed of POPC in the upper leaflet and either: (i) POPC:POPS (70:30), (ii) POPC:POPS:PI(4,5)P$_2$ (90:5:5), and (iii) POPC:POPS:PI(4,5)P$_2$ (80:10:10) in the lower leaflet using the CHARMM-GUI martini_maker module (*Jo et al., 2008*; *Qi et al., 2015*). The resulting systems were built using the Martini 2.2 force field topology and were later adapted to the Martini 3 (version m3.b3.2; *Souza and Marrink, 2020*) topology using the martinize2.py tool. For these systems, the PI(4,5)P$_2$ parameters were adapted from their Martini2.2 version by changing the names of the beads to the Martini3 naming scheme using as example other available lipids. These Martini3 PI(4,5)P$_2$ parameters are available in our github repository (see the Data avaliability section). Secondary structure restraints from the Martini force field were only applied to the TMD, and no harmonic bond restraints were defined in the building of these systems.

## All-atom models of JAK2-FERM-SH2 and its complex with PRLR-ICD$_{LID1}$

To build the JAK2-FERM-SH2+PRLR-ICD$_{K235-E284}$ complex the following structures where used: JAK1-FERM-SH2+IFNLR1 (PDB 5L04 [*Zhang et al., 2016*]), TYK2-FERM-SH2+IFNAR1 (PDB 4PO6 [*Wallweber et al., 2014*]), and JAK2-FERM-SH2 (PDB 4Z32 [*McNally et al., 2016*]). A structural alignment of the three FEMR-SH2 domains was performed with STAMP (*Russell and Barton, 1992*) using the Multiseq module (*Roberts et al., 2006*) of VMD (*Humphrey et al., 1996*). The model of PRLR-ICD$_{K235-E284}$ was generated with the MODELLER interface of Chimera using as template the aligned receptor-ICD regions present on the structures 5L04 and 4PO6. A total of 200 models were generated, and the best in terms of its DOPE score (*Shen and Sali, 2006*) was selected for further use. This resulted in a model of PRLR-ICD$_{K235-E284}$ bound to JAK2-FEMR-SH2. By combining this model with chain A of PDB 4Z32, a structural model of the JAK2-FERM-SH2+PRLR-ICD$_{K235-E284}$ complex was obtained. All-atom simulation systems were built for JAK2-FERM-SH2 and the JAK2-FERM-SH2+PRLR-ICD$_{K235-E284}$ complex model. The missing residues in the loop of F3 of JAK2-FERM-SH2 were completed using CHARMM-GUI pbd-reader module (*Jo et al., 2014*; *Jo et al., 2008*). Hydrogen atoms were automatically added to the protein using the psfgen plugin of VMD (*Humphrey et al., 1996*). Aspartate, glutamate, lysine, and arginine residues were charged, and histidine residues were neutral. Simulation boxes composed of solvent and 150 mM NaCl were generated using the CHARMM-GUI solution-builder module (*Jo et al., 2008*; *Lee et al., 2016*) using CHARMM36m (*Huang et al., 2017*) parameters and topologies for the protein and the TIP3P water model for the solvent.

## CG models of JAK2-FERM-SH2 and its complex with PRLR-ICD$_{LID1}$

To build CG models of JAK2-FERM-SH2 and complex between JAK2-FERM-SH2 and PRLR-ICD$_{K235-E284}$ complex, a conformation from their respective all-atom MD simulations of the complex was taken after 150 ns (see below). These conformations were used to generate a CG model using the martinize.py script. The Martini 2.2 force field (*de Jong et al., 2013*) was used, and intramolecular elastic bonds were defined for JAK2-FERM-SH2 in both systems. To keep the complex formed and to avoid a 'collapse' of the disordered PRLR-ICD$_{LID1}$, inter-molecular harmonic bonds were also defined between JAK-FERM-SH2 and PRLR-ICD$_{K235-E284}$ in the complex. In both cases, a force constant of 400 kJ mol$^{-1}$ nm$^{-2}$ and lower and upper elastic bond cutoffs of 5 Å and 9 Å, respectively, were used.

## CG models of JAK2-FERM-SH2 and JAK2-FERM-SH2 + PRLR-ICD$_{LID1}$ near a lipid bilayer

The relaxed CG-model of the JAK2-FERM-SH2+PRLR-ICD$_{LID1}$ complex or JAK2-FERM-SH2 alone (see below) was placed near (~7 Å) pre-equilibrated lipid bilayers with two different compositions: POPC on the upper leaflet and two different compositions on the lower leaflet: (i) POPC:POPS (70:30) and (ii) POPC:POPS:PI(4,5)P$_2$ (80:10:10). The systems were solvated with water beads and 150 mM NaCl. A total of 16 initial orientations of the protein were generated by rotating the protein around the x or the y axis (with z being the normal of the membrane).

## MD simulations

### CG MD simulations

CG MD simulations were performed with Gromacs 2016 or 2018 using the Martini 2.2 force field (*de Jong et al., 2013*) or the open beta version of the Martini 3 (3.b3.2) force field (*Souza and Marrink, 2020*). For the PRLR-TMD-ICD$_{K235-L284}$ simulations, we increased the strength of interactions between protein and water by 10% to avoid excessive compaction of the disordered regions, as has been previously done for IDPs and multi-domain proteins (*Kassem et al., 2021*; *Larsen et al., 2020*; *Thomasen et al., 2022*). Other simulation parameters, common to all the CG simulations performed, were chosen following the recommendations in *de Jong et al., 2016*. Briefly, a time step of 20 fs was used, and the Verlet cutoff scheme was used considering a buffer tolerance of 0.005 kJ/(mol ps atom). The reaction-field method was used for Coulomb interactions with a cutoff of 11 Å and a relative permittivity of $\varepsilon_r$ = 15. For van der Waals' interactions, a cutoff of 11 Å was used. The velocity rescaling thermostat was employed with a reference temperature of T=310 K, with a coupling constant of $\tau_T$ = 1 ps. For the equilibrations, the Berendsen barostat was employed ($P$=1 bar, $\tau_p$ = 3 ps), whereas the production runs were performed with a Parrinello-Rahman barostat ($P$=1 bar, $\tau_p$ = 12 ps). A semi-isotropic pressure coupling was used for all the systems that contained a lipid bilayer. For all systems, an initial round of equilibration with decreasing constraints applied to the protein beads and lipid beads was performed following the protocol provided by CHARMM-GUI Martini maker module. For the PRLR-TMD-ICD$_{K235-L284}$ simulations, a total of 11 μs of unconstrained MD were performed of which the first microsecond was considered as equilibration and the last 10 μs as production and used for analysis. For the JAK2-FERM-SH2 and the complex between JAK2-FERM-SH2 in solution, 1 μs of unconstrained simulation was performed. In the case of JAK2-FERM-SH2 or the complex between JAK2-FERM-SH2 and PRLR-TMD-ICD$_{K235-L284}$ near a lipid bilayer, an unconstrained run of 5 μs was performed for each system, and the complete trajectory was considered for analysis.

### All-atom MDs simulations

All-atom MD simulations were performed using GROMACS 2016 and 2018 (*Abraham et al., 2015*), using the CHARMM36m force field (*Huang et al., 2017*) for proteins and the TIP3P model for water. The initial system was minimized followed by position restrained simulation in two different phases, NVT and NPT. A 150 ns run of unconstrained NPT equilibration was then performed. The Berendsen thermostat was used for the constrained relaxation runs and the Nose-Hoover thermostat for the production runs. In all cases, the temperature was 310 K. For the NPT simulations, the Berendsen barostat was used during relaxations, and the Parinello-Rahman barostat was used in unconstrained production runs. In all cases, the target pressure was 1 atm. In all the simulations, the Verlet-cutoff scheme was used with a 2 fs timestep. A cutoff of 12 Å with a switching function starting at 10 Å was used for non-bonded interactions along with periodic boundary conditions. The Particle Mesh Ewald method was used to compute long-range electrostatic forces. Hydrogen atoms were constrained using the LINCS (*Hess et al., 1997*) algorithm.

### Trajectory analyses

Analysis of the obtained trajectories was performed using VMD plugins, GROMACS analysis tools, and in-house prepared tcl and python scripts, available on github (see below). For the characterization of the orientation of the JAK2-FERM-SH2+PRLR-ICD$_{K235-E284}$ complex with respect to the lipid bilayer, we used the geographical coordinate system with latitude and longitude devised by *Herzog et al., 2017*. Lipid densities were calculated with the Volmap plugin from VMD considering only the PO4 beads and a 1 Å grid. Density plots are shown as an enrichment score with values representing the percentage of enrichment or depletion with respect to the average value on the system as done previously in *Corradi et al., 2018*. All molecular renderings were done with VMD (*Humphrey et al., 1996*). Protein–protein contact maps were calculated using CONAN (*Mercadante et al., 2018*).

## Acknowledgements

The authors thank the SYNERGY, BRAINSTRUC, and REPIN consortia for valuable discussion and Signe A Sjørup and Katrine Franklin Mark for skilled technical assistance. We are grateful to Dr. Julie

Structural Biology and Molecular Biophysics

Schnipper for generating the AP1-2PH-PLCδ-GFP cells and to Prof. Vincent Goffin for initial discussions regarding the importance of the KxK-motifs in PRLR.

## Additional information

### Funding

| Funder | Grant reference number | Author |
|--------|------------------------|--------|
| Novo Nordisk Fonden | NNF18OC0033926 | Birthe B Kragelund |
| Novo Nordisk Fonden | NNF15OC0016670 | Birthe B Kragelund |
| Lundbeckfonden | BRAINSTRUC | Kresten Lindorff-Larsen |

The funders had no role in study design, data collection and interpretation, or the decision to submit the work for publication.

### Author contributions

Raul Araya-Secchi, Formal analysis, Validation, Investigation, Methodology, Writing - original draft; Katrine Bugge, Formal analysis, Validation, Investigation, Visualization, Methodology, Writing – review and editing; Pernille Seiffert, Formal analysis, Investigation, Visualization, Methodology, Writing - original draft; Amalie Petry, Gitte W Haxholm, Investigation; Kresten Lindorff-Larsen, Supervision, Funding acquisition, Validation, Writing – review and editing; Stine Falsig Pedersen, Lise Arleth, Conceptualization, Resources, Formal analysis, Supervision, Funding acquisition, Writing – review and editing; Birthe B Kragelund, Conceptualization, Resources, Formal analysis, Supervision, Funding acquisition, Validation, Visualization, Writing - original draft, Project administration, Writing – review and editing

### Author ORCIDs

Raul Araya-Secchi http://orcid.org/0000-0002-4872-3553
Katrine Bugge http://orcid.org/0000-0002-6286-6243
Pernille Seiffert http://orcid.org/0000-0003-4213-5336
Kresten Lindorff-Larsen http://orcid.org/0000-0002-4750-6039
Stine Falsig Pedersen http://orcid.org/0000-0002-3044-7714
Lise Arleth http://orcid.org/0000-0002-4694-4299
Birthe B Kragelund http://orcid.org/0000-0002-7454-1761

### Decision letter and Author response

Decision letter https://doi.org/10.7554/eLife.84645.sa1
Author response https://doi.org/10.7554/eLife.84645.sa2

## Additional files

### Supplementary files
• MDAR checklist

### Data availability

All data needed to evaluate the conclusions in the paper are present in the paper and/or the Supplementary Materials. The MD data and models together with the scripts used in the trajectory analysis are available on Github at https://github.com/Niels-Bohr-Institute-XNS-StructBiophys/PRLRmodel (copy archived at *Niels Bohr Institute-Structural Biophysics group, 2022*). NMR chemical shifts for the GHR-ICD-LID1 are deposited in the BioMagResBank under the accession number 51695.

The following dataset was generated:

| Author(s) | Year | Dataset title | Dataset URL | Database and Identifier |
|---|---|---|---|---|
| Kragelund BB, Haxholm GW, Seiffert P | 2023 | Chemical shift assignment of the intracellular domain of the prolactin receptor, residues 236-396 | https://doi.org/10.13018/BMR51695 | Biological Magnetic Resonance Data Bank, 10.13018/BMR51695 |

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
