## [Editor Report]

This important interdisciplinary study substantially advances our understanding of the prolactin receptor interactions with the membrane lipids and the effect of these interactions on cell signaling. The authors use a combination of state-of-the-art NMR structural analysis, simulations, and cellular assays to provide compelling experimental evidence for protein complexes being regulated by IDR-membrane interactions. The work will be of broad interest to structural biologists and biochemists, and the results presented herein are likely relevant for other non-tyrosine kinase receptors.

---

## [Decision Letter]

**Decision letter after peer review:**

Thank you for submitting your article "The prolactin receptor scaffolds Janus kinase 2 via co-structure formation with phosphoinositide-4,5-bisphosphate" for consideration by *eLife*. Your article has been reviewed by 3 peer reviewers, and the evaluation has been overseen by a Reviewing Editor and Amy Andreotti as the Senior Editor. The following individual involved in the review of your submission has agreed to reveal their identity: Frauke Gräter (Reviewer #2).

Essential revisions:

1) The introduction needs to be updated to include the motivation for focusing specifically on LID1. In addition, more detailed descriptions of the methodologies used need to be included, specifically how these have been previously successfully applied by researchers to study equivalent membrane systems.

2) Please provide a better connection between the JAK2-PRLR complex conformational states and their functional relevance. This is an important point, as the majority of the simulation part of the paper centers on suggesting different states of the PRLR-JAK2 complex, and their hypothesized functional relevance is presented as a major result, yet not verified by experiments.

3) The connection between simulations and mutational study is not very direct. It is not clear that the mutants can distinguish between the effects of PRLR-PIP2 interaction or PRLR-JAK2 interaction, yet this conclusion is still drawn from the data.

4) Please provide some evidence (from experiments or simulations) regarding the role of PIP3 interactions. Currently (very strong) experimental evidence is provided only for PIP2, showing it to be an important regulator, while no results are provided for PIP3, despite being included in the final model.

5) The conclusions drawn from the mutagenesis study (lines 547-555) are not directly supported by data. There is only a partial correlation between PRLR membrane localisation and STAT5 activation, and this is insufficient to attribute the unexplained part of the STAT5 activation to PRLR-JAK2 interactions without further studies.

6) Based on the method section, the JAK2-FERM-SH2 CG-MD simulations are based on Martini 2.2 forcefield. If it is the case, the results of the orientation of the protein towards the membrane may be affected, as there can be some underestimations of the aromatics-choline interactions (https://pubs.acs.org/doi/full/10.1021/acs.jctc.9b01194). This issue seems to be corrected in the new Martini3 version. Would it be possible to run a few control simulations using Martini3 to compare with Martini2.2 results and see if the protein orientation is affected? Otherwise, this limitation should at least be mentioned in the Discussion section.

7) Relatively large chemical shift changes are detected in PRLR res 285-290 upon lipid binding, these need to be discussed.

8) You might want to move some of the main figures to the supplementary data and further emphasize some of the major conclusions (for example Figure 3H and 3I).

*Reviewer #1 (Recommendations for the authors):*

The data of Figure 5A are explained as changes in peak intensities are the smallest for φ4G mutant. This is not true as K4G shows similar changes, and K4E shows even fewer changes.

Also, the explanation for the increase in intensity with titration to be due to weak binding is also not convincing. Could binding be measured by ITC for example to show indeed that one mutant binds weaker than the other?

*Reviewer #2 (Recommendations for the authors):*

The study would benefit from a clearer distinction of mutation effects on PRLR-PIP2 interaction or PRLR-JAK2 interaction by designing mutants that only affect one and not the other, if possible. The same applies to the JAK5 activation: experimental evidence is currently lacking.

Also, the connection between simulations and mutant experiments could be more direct:

– CIF motif identified to be involved in PIP2 interaction using NMR and simulation.

– KxK motif 1 (residue ~252) identified in simulation to interact with PIP2.

– KxK motif 2 (residue ~262) not identified in simulation or NMR as PIP2 contact but as a contact point with JAK2 in simulation (Figure 4 figure supplement 1).

– Parts of φ4G contact point with JAK2 in simulation but L252 is part of KxK motif 1.

Finally, the discussion of different conformational states needs to be revisited and refined: which state should be functionally rather switched off, which state switched on, and how is function inferred from the conformations?

Also, experiments/simulations on PIP3 or at least a discussion on how PIP3 interactions can be inferred from the results on PIP2 would strengthen the study (point 5 above).

*Reviewer #3 (Recommendations for the authors):*

There are a few points that need to be clarified:

1. In the introduction. While the biological context of this work is well explained, the methodological context is not really detailed. It gives the impression that technically speaking, this work is completely new, which is not the case. There are numerous published studies (both in terms of NMR and modelling) studying PIP2 lipids, IDPs, and membrane receptors. Thus, it may be interesting for the reader to see that other research has already been successfully applied to study equivalent membrane systems showing that the authors' strategy is indeed robust.

2. Figure 3 and Figure 3 supplement1: an RSMF analysis of the peptide LID1 would be useful to evaluate its degree of flexibility for each membrane system. It may help better understand how PIP2 lipids may partly structure the peptide.

3. From the method section, it is not clear where the PIP2 CG model is coming from. How this model can be compared with the recent parametrization of PIP2 (https://pubs.acs.org/doi/abs/10.1021/acs.jctc.1c00615)? If the two models are different how this can affect the modelling results?

4. My main concern is related to the JAK2-FERM-SH2 CG-MD simulations (Figure 4). Based on the method section these simulations seem to be based on Martini 2.2 forcefield. If it is the case, the results of the orientation of the protein towards the membrane may be affected as there can be some underestimations of the aromatics-choline interactions (https://pubs.acs.org/doi/full/10.1021/acs.jctc.9b01194). This issue seems to be corrected in the new Martini3 version. Would it be possible to run a few control simulations using Martini3 to compare with Martini2.2 results and see if the protein orientation is affected? Or, at least, mention this limitation in the Discussion section.

Typos and format:

p.6, figure2-A: MD simulations box: K335 needs to be changed to K235.

---

## [Author Response]

Essential revisions:1) The introduction needs to be updated to include the motivation for focusing specifically on LID1.

We have now provided our rationale for selectively focusing on the LID1 region in the PRLR. The selection was done to address how structural disorder in the juxtamembrane regions can transmit information on extracellular hormone binding to the intracellularly bound JAK2. This constitutes the first step in signaling on the intracellular side and—given the distance to the other two LIDs (LID2 and LID3) and their disconnect from the TMD by long disordered regions—LID1 was the focus. We have emphasized this choice in the introduction (p. 5) and in more detail in the result section (p. 5-6).

In addition, more detailed descriptions of the methodologies used need to be included, specifically how these have been previously successfully applied by researchers to study equivalent membrane systems.

We have included a paragraph in the introduction on previous work, which have successfully applied the used methodologies to study receptors and their interaction with lipids using integrative structural biology. We refer to various membrane systems of relevance and explain the rationale for choice of methodology for the study (p. 5).

2) Please provide a better connection between the JAK2-PRLR complex conformational states and their functional relevance. This is an important point, as the majority of the simulation part of the paper centers on suggesting different states of the PRLR-JAK2 complex, and their hypothesized functional relevance is presented as a major result, yet not verified by experiments.

In the original manuscript we already provided a detailed analysis of the different states, highlighting accessible residues and lipid interacting residues and comparing these across the states. From our experiments, including the cellular assays, we cannot with certainty link the two major states to active and/or inactive states, and therefore do not claim this in the manuscript. What we do put forward as a major result, is the presence of *more* than one major state as also stated in the abstract and in the conclusion of the result section as follows (p xx):

“Another key observation is the existence of different states in which different regions of both JAK2 FERM-SH2 domain and LID1 of PRLR are exposed to the solvent or hidden below the bilayer.”

In the discussion we do speculate as to which state may be the active and/or inactive dimer/monomer but we do not make conclusions. We have now made the finding of more states clearer in the text, and further compare the two major states, the Y and the Flat state, to the recent cryo-EM structures of JAK1 bound to IFNAR1, which lends some support to our speculations. The abstract now reads:

“We find that the co-structure exists in different states which we speculate could be relevant for turning signalling on and off.”

To discern the functional relevance of these state, if possible, will require experiments also in cells that by themselves would be a new study. We have to the best of our ability clarified that the functional relevance of the states has not been elucidated by the current work.

3) The connection between simulations and mutational study is not very direct. It is not clear that the mutants can distinguish between the effects of PRLR-PIP2 interaction or PRLR-JAK2 interaction, yet this conclusion is still drawn from the data.

This is a very important point, and we thank the reviewer for pointing out that we did not sufficiently clearly describe our choices and the connection between the experimental data, the simulations, and the mutational study. As highlighted by Reviewer 2, the experimental data have provided a guideline for the selection, however, the arguments for choosing the mutations were unfortunately omitted from the text.

We have now explained these points in more detail, both regarding the choices (p. 16,17) and the data analysis and conclusions (p. 19,20). Furthermore, we have emphasized that in the case of PRLR, where a co-structure is formed, separating functions by mutation may be complicated by the intimate interplay of the structural preferences among the three components.

4) Please provide some evidence (from experiments or simulations) regarding the role of PIP3 interactions. Currently (very strong) experimental evidence is provided only for PIP2, showing it to be an important regulator, while no results are provided for PIP3, despite being included in the final model.

In our previous work addressing the affinity for different phosphoinositides using lipid dot blots we observed a preference for certain species, including PI(4,5)P2 (Haxholm et al., BJ, 2015). In that study, we also observed that there was no detectable affinity for PI(3,4,5)P3. We have now more explicitly described these data, both in the introduction (p.4), in the result section (p. 8) and later in the discussion (p. 21). We thank the reviewer for bringing this up.

5) The conclusions drawn from the mutagenesis study (lines 547-555) are not directly supported by data. There is only a partial correlation between PRLR membrane localisation and STAT5 activation, and this is insufficient to attribute the unexplained part of the STAT5 activation to PRLR-JAK2 interactions without further studies.

We agree and have toned down the conclusion. In conjunction with the more in-depth clarification of the results in this section, the conclusions should now more clearly reflect the data.

6) Based on the method section, the JAK2-FERM-SH2 CG-MD simulations are based on Martini 2.2 forcefield. If it is the case, the results of the orientation of the protein towards the membrane may be affected, as there can be some underestimations of the aromatics-choline interactions (https://pubs.acs.org/doi/full/10.1021/acs.jctc.9b01194). This issue seems to be corrected in the new Martini3 version. Would it be possible to run a few control simulations using Martini3 to compare with Martini2.2 results and see if the protein orientation is affected? Otherwise, this limitation should at least be mentioned in the Discussion section.

We thank the reviewer for the insightful comment and acknowledge the potential limitation of using the Martini2.2 forcefield in our simulations. As the reviewer mentions, the Martini2.2 forcefield may underestimate the aromatic-choline interactions, which could affect the orientation of the protein towards the membrane. While we did not directly address this issue in our study, we note that the main protein – lipid interactions we observe are driven by electrostatic interactions between positively charged residues (K and R) and the PI(4,5)P_2_ headgroups (and POPS in the systems without PI(4,5)P_2_) which could mitigate the impact of the underestimation of cation-π interactions. As suggested by the reviewer, we have included a discussion of these limitations in the revised manuscript (p. 14).

7) Relatively large chemical shift changes are detected in PRLR res 285-290 upon lipid binding, these need to be discussed.

The region referred to contributes to binding but is on the edge of the main binding site and where the local affinities are weaker. Therefore, the exchange rate is high and allows for following the chemical shift changes. In support of this, we see an almost inverse correlation between the CSPs and the changes in intensities. For the main binding site, the exchange rate between bound and free states is slower because the affinity is stronger. Therefore, we cannot follow the chemical shifts to extract the CSPs to the bound state, as the peaks disappear. We have commented on this in the main text (p.8) as follows:

“In the region from D285-E292 we observed an almost inverse correlation between the CSPs and the intensities. This suggests that in contrast to the preceding region, a faster local exchange rate allows us to follow the resonances from the bound state in this region, giving rise to the large CSPs.”

8) You might want to move some of the main figures to the supplementary data and further emphasize some of the major conclusions (for example Figure 3H and 3I).

A new version of Figure 3 has been generated to consider the reviewers’ comments and suggestions. This figure has been restructured to further emphasize some of the major conclusions obtained from the simulations. We have moved the former Figure 3 A, B, C and D to the supplemental information to increase the focus. Thus, the former Figure 3H is now Figure 3C and it has been significantly enlarged and include some corrections and edits: There was an error in the script we used to process the volumetric data containing the density (from VMD’s Volmap tool). Thus, the -150% issue was solved. Also, we noticed that since Volmap is not aware of the periodic boundary conditions (PBC) of the system, the edges of the box will contain low density values due to fluctuations in the number of lipids in this zone. That leads to average values near 0 which result in an abnormal depletion % at the edges observed in the previous figure. This has been solved by filtering out those values and showing the density only in the regions further inside the simulation box.

The former Figure 3G (now Figure 3C) now also includes residues up to H300 as suggested. We thank the reviewers for suggesting these improvements.

Reviewer #1 (Recommendations for the authors):The data of Figure 5A are explained as changes in peak intensities are the smallest for φ4G mutant. This is not true as K4G shows similar changes, and K4E shows even fewer changes.Also, the explanation for the increase in intensity with titration to be due to weak binding is also not convincing. Could binding be measured by ITC for example to show indeed that one mutant binds weaker than the other?

We thank the reviewer for bringing this mistake to our attention. In our analyses, we compared both the intensity changes observed for the variant (internal comparison) and with those observed for the wild-type (across variants) and these were mixed up in the final text. We have now corrected the text accordingly and fully agree with the reviewer’s analyses.

Regarding a more quantitative measure of the affinities, ITC would require an affinity at least in the μM range, and since the complex is stabilized further with the membrane surface as well as by binding JAK2, it would not be straightforward.

Reviewer #2 (Recommendations for the authors):The study would benefit from a clearer distinction of mutation effects on PRLR-PIP2 interaction or PRLR-JAK2 interaction by designing mutants that only affect one and not the other, if possible. The same applies to the JAK5 activation: experimental evidence is currently lacking.Also, the connection between simulations and mutant experiments could be more direct:– CIF motif identified to be involved in PIP2 interaction using NMR and simulation.– KxK motif 1 (residue ~252) identified in simulation to interact with PIP2.– KxK motif 2 (residue ~262) not identified in simulation or NMR as PIP2 contact but as a contact point with JAK2 in simulation (Figure 4 figure supplement 1).– Parts of φ4G contact point with JAK2 in simulation but L252 is part of KxK motif 1.

This is a very important point, and we are grateful for the reviewer to point out that we did not sufficiently clearly describe the background for our choices. As highlighted by Reviewer 2, and inserted above, the experimental data have provided a guideline for the selection, however, this was completely omitted from the text. Combined with the confusion on the evaluation on parts of the variant data, mentioned by reviewer 1, this made the conclusion from these data non-transparent. We have explained in much more detail, both on the choices (p. 16,17) and on the data analysis and conclusions (p. 19,20). Furthermore, we have emphasized that the separation of function by mutation may be complicated by the intimate interplay of the structural preferences among the three components of the co-structure.

Finally, the discussion of different conformational states needs to be revisited and refined: which state should be functionally rather switched off, which state switched on, and how is function inferred from the conformations?Also, experiments/simulations on PIP3 or at least a discussion on how PIP3 interactions can be inferred from the results on PIP2 would strengthen the study (point 5 above).

In the original manuscript we already provided a detailed analysis of the different states, highlighting accessible residues and lipid interacting residues and compare these across the states. From our experiment, including those performed in cellular assay, we cannot with certainty link the two major state to active and/or inactive states. We have therefore no intention or support from the data to claim this. However, what we do put forward as a major result, in the presence of *more* than one major state as also stated in the abstract and in the conclusion of the result section:

“Another key observation is the existence of different states in which different regions of both JAK2 FERM-SH2 domain and LID1 of PRLR are exposed to the solvent or hidden below the bilayer.”

In the discussion we do speculate as to which state may be the active and/or inactive dimer/monomer but make no firm claims. We have now made the major find of more states clearer in the text, and further compare the two major states, the Y and the Flat state, to the recent cryo-EM structures of JAK1 bound to IFNAR1, which lend some support to our speculations. The abstract now reads:

“We find that the co-structure exists in different states which we speculate could be relevant for turning signalling on and off.”

To discern the functional relevance of these state, if possible, will require experiments also in cells that by themselves would be a new study. We have to the be best of our ability clarified that the functional relevance of the states has not been clarified by the current work.

Reviewer #3 (Recommendations for the authors):There are a few points that need to be clarified:1. In the introduction. While the biological context of this work is well explained, the methodological context is not really detailed. It gives the impression that technically speaking, this work is completely new, which is not the case. There are numerous published studies (both in terms of NMR and modelling) studying PIP2 lipids, IDPs, and membrane receptors. Thus, it may be interesting for the reader to see that other research has already been successfully applied to study equivalent membrane systems showing that the authors' strategy is indeed robust.

We have included a paragraph in the introduction on previous work which have successfully applied the used methodologies to study receptors and their interaction with lipids using integrative structural biology. We refer here to various membrane systems of relevance and explain the methodological background for the study (p. 5).

2. Figure 3 and Figure 3 supplement1: an RSMF analysis of the peptide LID1 would be useful to evaluate its degree of flexibility for each membrane system. It may help better understand how PIP2 lipids may partly structure the peptide.

PRLR-TMD-LID1 residues simulated with PI(4,5)P2 and POPC lipids on the lower leaflet to evaluate if PI(4,5)P2 affects the flexibility of the region of the protein that interact with it (Figure 3 – supplemental figure 1C). What we found is that apparently the interactions (with penetration) of the hydrophobic residues of the ICJM and BOX1 stabilize and reduce the flexibility in this region, but it is not necessarily PIP2 and its interaction with the positively charged residues that causes this, as the same effect is observed wit POPS. Thus, this is strengthening our suggestion that the

“The stabilization of the structure provided by the hydrophobic residues from the ICJM and BOX1 is also reflected on their decrease in flexibility, observed as a shoulder on the RMSF-BB plot, for the residues that comprise the ICJM and BOX1 of PRLR-LID1. Very similar profiles of the RMSF-BB plot was obtained for the systems with respectively 5 and 10% PI(4,5)P_2_ in POPC:POPS, suggesting that the loss in flexibility is coupled to the buried hydrophobic residues rather than to specific PI(4,5)P_2_ interaction (Figure 3 —figure supplement 2C”).

We discuss these data on p. 10.

3. From the method section, it is not clear where the PIP2 CG model is coming from. How this model can be compared with the recent parametrization of PIP2 (https://pubs.acs.org/doi/abs/10.1021/acs.jctc.1c00615)? If the two models are different how this can affect the modelling results?

We thank the reviewer for pointing this out. Indeed, we did not clearly mention the origin of our Martini 3 parameters for PIP2. At the time when we performed the simulations the only available version of M3 was m3.b3.2 and there were no PIP2 parameters available yet. So, we adapted the PIP2 parameters from the martini2.2 forcefield. These parameters are available on our github and zeonodo repositories for this project. With respect to the new martini 3 parameters for PIP2 referred by the reviewer, it is difficult to compare with our parameters as the reported improvements revolve around self-interaction of phosphoinositide and their aggregation in the presence of ca^2+^. With respect to protein-PIP2 interactions, the authors only show that the Martini 3 PI(4,5)P2 model is correctly interacting with proteins and replicating experimental findings, but no direct comparison with Martini2 is made.

On the other hand, it is known that the martini2 PIP2 model was able to correctly predict the location of PIP2 binding sites on Kir channels (https://doi.org/10.1021/bi9013193) among other successful examples (10.1039/C3CS60093A). Thus, we don’t expect that the use of the parameters derived from the Martini 2 model affect our results, however, further studies of this systems should take advantage of the recently published new models. A clarification regarding the precedence of the M3 PIP2 parameters was included in the methods section.

4. My main concern is related to the JAK2-FERM-SH2 CG-MD simulations (Figure 4). Based on the method section these simulations seem to be based on Martini 2.2 forcefield. If it is the case, the results of the orientation of the protein towards the membrane may be affected as there can be some underestimations of the aromatics-choline interactions (https://pubs.acs.org/doi/full/10.1021/acs.jctc.9b01194). This issue seems to be corrected in the new Martini3 version. Would it be possible to run a few control simulations using Martini3 to compare with Martini2.2 results and see if the protein orientation is affected? Or, at least, mention this limitation in the Discussion section.

We thank the reviewer for the insightful comment and acknowledge the potential limitation of using the Martini2.2 forcefield in our simulations. As the reviewer mentions, the Martini2.2 forcefield may underestimate the aromatic-choline interactions, which could affect the orientation of the protein towards the membrane. While we did not directly address this issue in our study, we do observe that the main protein – lipid interactions in these simulations are driven by electrostatic interactions between positively charged residues (K and R) and the PI(4,5)P_2_ headgroups (and POPS in the systems without PI(4,5)P_2_) which could mitigate the impact of the underestimation of cation-π interactions. We have included a discussion of this limitation in the revised manuscript (p. 14).

Typos and format:p.6, figure2-A: MD simulations box: K335 needs to be changed to K235.

This has been corrected.